

# A fast algorithm to compute a curve of confidence upper bounds for the False Discovery Proportion using a reference family with a forest structure

ISSN 2824-7795

Guillermo Durand [1]    Laboratoire de Mathématiques d'Orsay, Université Paris-Saclay

Date published: 2025-01-31    Last modified: 2025-01-31

## Abstract

This paper presents a new algorithm (and an additional trick) that allows to compute fastly an entire curve of post hoc bounds for the False Discovery Proportion when the underlying bound $V^*_{\mathfrak{R}}$ construction is based on a reference family $\mathfrak{R}$ with a forest structure à la Durand et al. (2020). By an entire curve, we mean the values $V^*_{\mathfrak{R}}(S_1), \ldots, V^*_{\mathfrak{R}}(S_m)$ computed on a path of increasing selection sets $S_1 \subsetneq \cdots \subsetneq S_m$, $|S_t| = t$. The new algorithm leverages the fact that going from $S_t$ to $S_{t+1}$ is done by adding only one hypothesis.

*Keywords:* multiple testing, algorithmic, post hoc inference, false discovery proportion, confidence bound

# Contents

[1] Corresponding author: guillermo.durand@universite-paris-saclay.fr

# 1 Introduction

Multiple testing theory is often used for exploratory analysis, like Genome-Wide Association Studies, where multiple features are tested to find promising ones. Classical multiple testing theory like Family-Wise Error Rate (FWER) control or False Discovery Rate (FDR) control (Benjamini and Hochberg, 1995) can be used, but a more recent trend consists in the computation of post hoc bounds, also named post selection bounds or confidence envelopes, for the number of false positives, or, equivalently, for the False Discovery Proportion (FDP). This approach is notably advocated for in the context of exploratory research by (Goeman and Solari, 2011, Section 1).

Mathematically speaking, a confidence upper bound (we prefer to say upper bound instead of envelope for obvious reasons) is a function $\hat{V} : \mathscr{P}(\mathbb{N}_m^*) \to \mathbb{N}_m$, where $\mathbb{N}_m = \{0, \dots, m\}$, $\mathbb{N}_m^* = \{1, \dots, m\}$ and $m$ is the number of hypotheses, such that

$$\forall \alpha \in ]0, 1[, \mathbb{P}\left(\forall S \subseteq \mathbb{N}_m^*, |S \cap \mathscr{H}_0| \leq \hat{V}(S)\right) \geq 1 - \alpha. \tag{1}$$

Here, $\alpha$ is a target error rate and $\mathscr{H}_0$ is the set of hypotheses indices that are true null hypotheses. Note that the construction of $\hat{V}$ depends on $\alpha$ and on the random data $X$ and the dependence is omitted to lighten notation and because there is no ambiguity. The meaning of Equation (1) is that $\hat{V}$ provides an upper bound of the number of null hypotheses in $S$ for any selection set $S \subseteq \mathbb{N}_m^*$, which allows the user to perform post hoc selection on their data without breaching the statistical guarantee. Also note that by dividing by $|S| \vee 1$ in Equation (1) we also get a confidence bound for the FDP:

$$\forall \alpha \in ]0, 1[, \mathbb{P}\left(\forall S \subseteq \mathbb{N}_m^*, \mathrm{FDP}(S) \leq \frac{\hat{V}(S)}{|S| \vee 1}\right) \geq 1 - \alpha. \tag{2}$$

So post hoc bounds provide ways to construct FDP-controlling sets instead of FDR-controlling sets, which is much more desirable given the nature of the FDR as an expected value. See for example (Bogdan et al., 2015, Figure 4) for a credible example where the FDR is controlled but the FDP has a highly undesirable behavior (either 0 because no discoveries at all are made, either higher than the target level).

The first confidence bounds are found in Genovese and Wasserman (2006) and Meinshausen (2006), although, in the latter, only for selection sets of the form $\{i \in \mathbb{N}_m : P_i \leq t\}$ where $P_i$ is the $p$-value associated to the null hypothesis $H_{0,i}$. In Goeman and Solari (2011) the authors re-wrote the generic construction of Genovese and Wasserman (2006) in terms of closed testing Marcus et al. (1976), proposed several practical constructions and sparked a new interest in multiple testing procedures based on confidence envelopes. This work was followed by a prolific series of works like Meijer et al. (2015) or Vesely et al. (2023). In Blanchard et al. (2020), the authors introduce the new point of view of references families (see Section 2.2) to construct post hoc bounds, and show the links between this meta-technique and the closed testing one, along with new bounds.

Following the reference family trail, in Durand et al. (2020) the authors introduce new reference families with a special set-theoretic constraint that allows an efficient computation of the bound denoted by $V_{\mathfrak{R}}^*$ on a single selection set $S$. The problem is that one often wants to compute $V_{\mathfrak{R}}^*$ on a whole path of selection sets $(S_t)_{t \in \mathbb{N}_m^*}$, for example the hypotheses attached to the $t$ smallest $p$-values. Whereas the algorithm provided in the aforementioned work (Durand et al., 2020, Algorithm 1), which is reproduced here, see Algorithm 1 , is fast for a single evaluation, it is slow and inefficient to repeatedly call it to compute each $V_{\mathfrak{R}}^*(S_t)$. If the $S_t$'s are nested, and growing by one, that is $S_1 \subsetneq \cdots \subseteq S_m$ and $|S_t| = t$, there is a way to efficiently compute $\left(V_{\mathfrak{R}}^*(S_t)\right)_{t \in \mathbb{N}_m}$ by leveraging the nested structure.

This is the main contribution of the present paper: a new and fast algorithm computing the curve $\left(V_{\mathfrak{R}}^*(S_t)\right)_{t \in \mathbb{N}_m}$ for a nested path of selection sets, that is presented in Section 3.2. An additional

algorithm that can speed up computations both for the single-evaluation algorithm and the new curve-evaluation algorithm is also presented, in Section 3.1. A detailed example illustrating how the new algorithms work is provided in Section 3.3, and the proof that the fast algorithm indeed computes correctly the curve is in Section 3.4. In Section 2.1, all necessary notation and vocabulary is re-introduced, most of it being the same as in Durand et al. (2020). In Section 4 we discuss the current implementations of all the presented algorithms in the R (R Core Team, 2024) package sanssouci (Neuvial et al., 2024). Finally, a few numerical experiments are presented in Section Section 5 to demonstrate the computation time gain.

## 2 Notation and reference family methodology

### 2.1 Multiple testing notation

As is usual in multiple testing theory, we consider a probability space $(\Omega, \mathscr{A}, \mathbb{P})$, a model $\mathscr{P}$ on a measurable space $(\mathscr{X}, \mathfrak{X})$, and data that is represented by a random variable $X : (\Omega, \mathscr{A}) \to (\mathscr{X}, \mathfrak{X})$ with $X \sim P \in \mathscr{P}$, that is, the law of $X$ is comprised in the model $\mathscr{P}$.

Then we consider $m \geq 1$ null hypotheses $H_{0,1}, \ldots, H_{0,m}$ which formally are submodels, that is subsets of $\mathscr{P}$. The associated alternative hypotheses $H_{1,1}, \ldots, H_{1,m}$ are submodels such that $H_{0,i} \cap H_{1,i} = \emptyset$ for all $i \in \mathbb{N}_m^*$. We denote by $\mathscr{H}_0 = \mathscr{H}_0(P)$ (the dependence in $P$ will be dropped when there is no ambiguity) the set of all null hypotheses that are true, that is $\mathscr{H}_0(P) = \{i \in \mathbb{N}_m^* : P \in H_{0,i}\}$. In other words, $H_{0,i}$ is true if and only if $i \in \mathscr{H}_0$. For testing each $H_{0,i}, i \in \mathbb{N}_m^*$, we have at hand a $p$-value $p_i = p_i(X)$ (the dependence in $X$ will be dropped when there is no ambiguity) which is a random variable with the following property : if $i \in \mathscr{H}_0$, then the law of $p_i$ is super-uniform, which is sometimes denoted $\mathscr{L}(p_i) \succeq \mathscr{U}([0,1])$. This means that in such case, the cumulative distribution function (cdf) of $p_i$ is always smaller than or equal to the cdf of a random variable $U \sim \mathscr{U}([0,1])$ :

$$\forall x \in \mathbb{R}, \mathbb{P}\left(p_i \leq x\right) \leq \mathbb{P}\left(U \leq x\right) = 0 \vee (x \wedge 1). \tag{3}$$

For every subset of hypotheses $S \subseteq \mathbb{N}_m^*$, let $V(S) = |S \cap \mathscr{H}_0|$. If we think of $S$ as a selection set of hypotheses deemed significant, $V(S)$ is then the number of false positives (FP) in $S$. $V(S)$ is our main object of interest and the quantity that we wish to over-estimate with confidence upper bounds (see Equation (1) or the more formal Equation (4) below).

Finally let us consider the following toy example, that will be re-used in the remainder of the paper.

**Example 2.1** (Gaussian one-sided). In this case we assume that $X = (X_1, \ldots, X_m)$ is a Gaussian vector and the null hypotheses refer to the nullity of the means in contrast to their positivity. That is, formally, $(\mathscr{X}, \mathfrak{X}) = (\mathbb{R}^m, \mathscr{B}(\mathbb{R}^m))$, $\mathscr{P} = \{\mathscr{N}(\boldsymbol{\mu}, \Sigma) : \forall j \in \mathbb{N}_m^*, \mu_j \geq 0, \Sigma \text{ positive semidefinite}\}$, for each $i \in \mathbb{N}_m^*$, $H_{0,i} = \{\mathscr{N}(\boldsymbol{\mu}, \Sigma) \in \mathscr{P} : \mu_i = 0\}$ and $H_{1,i} = \{\mathscr{N}(\boldsymbol{\mu}, \Sigma) \in \mathscr{P} : \mu_i > 0\}$. Then we can construct $p$-values by letting $p_i(X) = \bar{\Phi}(X_i) = 1 - \Phi(X_i)$, where $\Phi$ denotes the cdf of $\mathscr{N}(0,1)$ and $\bar{\Phi}$ the associated survival function.

### 2.2 Post hoc bounds with reference families

With the formalism introduced in last section, a confidence upper bound is a functional $\hat{V} : \mathscr{X} \times ]0, 1[ \to (\mathscr{P}(\mathbb{N}_m^*) \to \mathbb{N}_m)$ such that,

$$\forall P \in \mathscr{P}, \forall X \sim P, \forall \alpha \in ]0, 1[, \mathbb{P}\left(\forall S \subseteq \mathbb{N}_m^*, V(S) \leq \hat{V}(X, \alpha)(S)\right) \geq 1 - \alpha. \tag{4}$$

In the remainder, the dependence in $(X, \alpha)$ will be dropped when there is no ambiguity and $\hat{V}(X, \alpha)$ will simply be written $\hat{V}$.

As said in the Introduction, many constructions, ultimately theoretically equivalent but differing by the practical steps involved, exist, and in this paper we focus on the meta-construction of Blanchard et al. (2020) based on reference families. A reference family is a family $\mathfrak{R} = \mathfrak{R}(X, \alpha) = (R_k, \zeta_k)_{k \in \mathcal{K}}$ with $|\mathcal{K}| \leq 2^m$, $R_k \subseteq \mathbb{N}_m^*$, $\zeta_k \in \{0, \dots, |R_k|\}$ where everything (that is, $\mathcal{K}$ and all the $R_k$ and $\zeta_k$) depends on $(X, \alpha)$ but the dependency is not explicitly written. The $R_k$ are all distinct. We also define the following error criterion for a reference family, named Joint Error Rate (JER):

$$\mathrm{JER}(\mathfrak{R}) = \mathbb{P}\left(\exists k \in \mathcal{K}, |R_k \cap \mathcal{H}_0| > \zeta_k\right) = \mathbb{P}\left(\exists k \in \mathcal{K}, V(R_k) > \zeta_k\right). \tag{5}$$

In the following, we are only interested in reference families that control the JER at level $\alpha$:

$$\forall P \in \mathscr{P}, \forall X \sim P, \forall \alpha \in ]0, 1[, 1 - \mathrm{JER}(\mathfrak{R}(X, \alpha)) = \mathbb{P}\left(\forall k \in \mathcal{K}, V(R_k) \leq \zeta_k\right) \geq 1 - \alpha. \tag{6}$$

Note that Equation (6) is really similar to Equation (4) except that the uniform guarantee, instead of being over all $S \subseteq \mathbb{N}_m^*$, is only over all the $R_k \subseteq \mathbb{N}_m^*, k \in \mathcal{K}$, with $\mathcal{K}$ having cardinality potentially much smaller than $2^m$. A global confidence bound is then derived from a JER-controlling reference family by interpolation. Let

$$\mathscr{A}(\mathfrak{R}) = \{A \subseteq \mathbb{N}_m^* : \forall k \in \mathcal{K}, |R_k \cap A| \leq \zeta_k\}. \tag{7}$$

What says the JER control is that $\mathcal{H}_0 \in \mathscr{A}(\mathfrak{R})$. We leverage this information with the following confidence bound construction:

$$V_{\mathfrak{R}}^*(S) = \max_{A \in \mathscr{A}(\mathfrak{R})} |S \cap A| \tag{8}$$

which optimally uses the information provided by the JER control of the reference family, as proven by Proposition 2.1 of Blanchard et al. (2020). Because of the $\max_{A \in \mathscr{A}(\mathfrak{R})}$, the computation of $V_{\mathfrak{R}}^*(S)$ is generally intractable (see Proposition 2.2 of Blanchard et al. (2020)), but for specific structures of reference families, a polynomial computation can be derived. This is the topic of Durand et al. (2020) and of next section.

## 2.3 Regions with a forest structure

The core concept of this section is to assume that the regions $R_k$'s of the reference family are what we called in Durand et al. (2020) a forest structure, that is two regions are either disjoint or nested:

$$\forall k, k' \in \mathcal{K}, R_k \cap R_{k'} \in \{R_k, R_{k'}, \varnothing\}. \tag{9}$$

Representing the $R_k$'s with a directed graph, where there is an oriented edge $R_k \leftarrow R_{k'}$ if and only if $R_k \subset R_{k'}$ and there is no $R_{k''}$ such that $R_k \subsetneq R_{k''} \subsetneq R_{k'}$ gives a forest, hence the name. See Example 2.2 and its representation in Figure 1.

We also need to introduce the notion of depth with the following function:

$$\phi : \begin{cases} \mathcal{K} & \to & \mathbb{N}^* \\ k & \mapsto & 1 + |\{k' \in \mathcal{K} : R_k \subsetneq R_{k'}\}|. \end{cases} \tag{10}$$

**Example 2.2.** Let $m = 25$, $R_1 = \{1, \dots, 20\}$, $R_2 = \{1, 2\}$, $R_3 = \{3, \dots, 10\}$, $R_4 = \{11, \dots, 20\}$, $R_5 = \{5, \dots, 10\}$, $R_6 = \{11, \dots, 16\}$, $R_7 = \{17, \dots, 20\}$, $R_8 = \{21, 22\}$, $R_9 = \{22\}$. This is the same example as Example 2 of Durand et al. (2020) and it is graphically depicted in Figure 1. The sets $R_1$, $R_8$ are of depth 1; the sets $R_2, R_3, R_4, R_9$ are of depth 2; the sets $R_5, R_6, R_7$ are of depth 3.

Another tool of Durand et al. (2020) that will be used is its Lemma 2, that is the identification of $\mathfrak{R}$ with a set $\mathscr{C} \subset \left\{(i, j) \in \left(\mathbb{N}_N^*\right)^2 : i \leq j\right\}$ such that for $(i, j), (i', j') \in \mathscr{C}, \{i, \dots, j\} \cap \{i', \dots, j'\} \in$

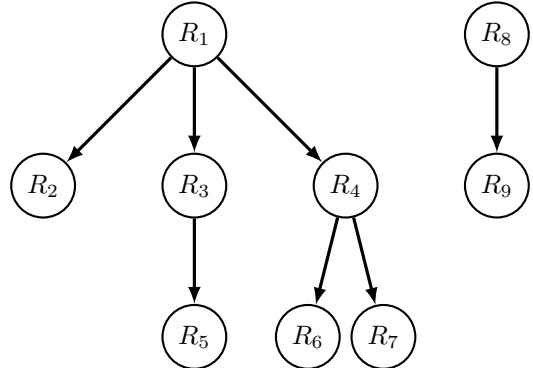

Figure 1: The regions of Example 2.2.

$\{\varnothing, \{i, \dots, j\}, \{i', \dots j'\}\}$. With this identification, each $R_k = R_{(i,j)}$ can be written as $P_{i:j} = \bigcup_{i \leq n \leq j} P_n$ where $(P_n)_{1 \leq n \leq N}$ is a partition of $\mathbb{N}_m^*$. The $P_n$'s were called atoms in Durand et al. (2020) because they have the thinnest granularity in the structure, but to continue the analogy with graphs, forests and trees, they can also be called leafs. See Example 2.3 for a concrete example.

**Example 2.3** (Continuation of Example 2.2). For the reference family given in Example 2.2, a partition of atoms is given by $P_1 = R_2$, $P_2 = R_3 \setminus R_5$, $P_3 = R_5$, $P_4 = R_6$, $P_5 = R_7$, $P_6 = R_8 \setminus R_9$, $P_7 = R_9$, $P_8 = \mathbb{N}_m^* \setminus \{R_1 \cup R_8\}$. Then $R_1 = P_{1:5}$, $R_3 = P_{2:3}$, $R_4 = P_{4:5}$ and $R_8 = P_{6:7}$. Note that not all atoms are regions of the family. Those new labels are graphically depicted in Figure 2. The nodes that correspond to atoms that are not in the family are depicted with a dashed circle, and all atoms are depicted in gray. This is the same example as Example 3 of Durand et al. (2020).

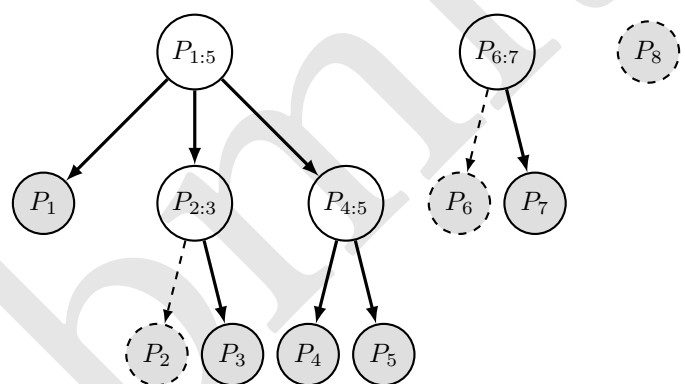

Figure 2: The regions of Example 2.2 but with the labels of Example 2.3.

When all leaves are regions of the family, it is said that the family is complete. If this is not the case, the family can easily be completed by adding the missing leaves (and using their cardinality as associated $\zeta$) without changing the value $V_{\mathfrak{R}}^*$. See Definition 2, Lemma 6 and Algorithm 2 of Durand et al. (2020) for the details.

Durand et al. (2020) also proved in their Theorem 1 that:

$$V_{\mathfrak{R}}^*(S) = \min_{Q \subseteq \mathscr{K}} \left( \sum_{k' \in Q} \zeta_{k'} \wedge |S \cap R_{k'}| + \left| S \setminus \bigcup_{k' \in Q} R_{k'} \right| \right) \tag{11}$$

and, even better, in their Corollary 1 *(iii)* that:

$$V_{\mathfrak{R}}^*(S) = \min_{Q \in \mathfrak{P}} \sum_{k' \in Q} \zeta_{k'} \wedge |S \cap R_{k'}|, \tag{12}$$

147 provided that the family is complete. Here, $\mathfrak{P} \subseteq \mathscr{P}(\mathscr{K})$ is the set of subsets of $\mathscr{K}$ that realize a
148 partition, that is, the set of $Q \subseteq \mathscr{K}$ such that the $R_k$, $k \in Q$, form a partition of $\mathbb{N}_m^*$. So the minimum
149 in Equation (12) is over way less elements than in Equation (11).

150 Finally, that paper provides a polynomial algorithm to $V_{\mathfrak{R}}^*(S)$ for a single $S \subseteq \mathbb{N}_m^*$, which we reproduce
151 here in Algorithm 1 . The family is assumed complete, otherwise the first step would be to complete
152 it. In the original paper, $\mathscr{K}^h$ used to designate the elements of $\mathscr{K}$ at depth $h$ plus the atoms at depth
153 $\leq h$. Actually one can realize that the last assumption is not needed for this algorithm to perform
154 exactly the same, with the added benefit of not repeating computations at the atoms that don't have
155 the maximal depth. The only change is that sometimes $Succ_k$ can be empty, in which case we simply
156 let $newVec_k = \zeta_k \wedge |S \cap R_k|$. Thus, here in this paper, we define $\mathscr{K}^h$ as only the elements of $\mathscr{K}$ at
157 depth $h$ (the previous intricate definition may still be necessary for the proof of Theorem 1 of Durand
158 et al. (2020)): $\mathscr{K}^h = \{(i,j) \in \mathscr{K} : \phi(i,j) = h\}$, $h \geq 1$. This is the only deviation from the notation of
159 Durand et al. (2020). Finally note that in the ongoing analogy with graph theory, the elements of $\mathscr{K}^1$
160 are the roots of the different trees making up the forest.

---

**Algorithm 1** Computation of a given $V_{\mathfrak{R}}^*(S)$

---

1: **procedure** VSTAR(S, $\mathfrak{R} = (R_k, \zeta_k)_{k \in \mathscr{K}}$ with $\mathfrak{R}$ complete)
2:      $H \leftarrow \max_{k \in \mathscr{K}} \phi(k)$          ▷ maximum depth
3:      **for** $h = H - 1, \dots, 1$ **do**
4:          $\mathscr{K}^h \leftarrow \{k \in \mathscr{K} : \phi(k) = h\}$
5:          $newVec \leftarrow (0)_{k \in \mathscr{K}^h}$
6:          **for** $k \in \mathscr{K}^h$ **do**
7:              $Succ_k \leftarrow \{k' \in \mathscr{K}^{h+1} : R_{k'} \subseteq R_k\}$
8:              **if** $Succ_k = \varnothing$ **then**
9:                  $newVec_k \leftarrow \zeta_k \wedge |S \cap R_k|$
10:              **else**
11:                  $newVec_k \leftarrow \min\left(\zeta_k \wedge |S \cap R_k|, \sum_{k' \in Succ_k} Vec_{k'}\right)$
12:              **end if**
13:          **end for**
14:          $Vec \leftarrow newVec$
15:      **end for**
16:      **return** $\sum_{k \in \mathscr{K}^1} Vec_k$
17: **end procedure**

---

> 💡 Tip
>
> In the practical implementation of this algorithm (and of the following Algorithm 2 ), $Vec$ and
> $newVec$ are always of size $N$ (the number of leaves) instead of the cardinality of $\mathscr{K}^h$. And the sum
> $\sum_{k' \in Succ_k} Vec_{k'}$ is really easy to compute: if $R_k = R_{(i_0, i_p - 1)} = \bigcup_{j=1}^{p} R_{(i_{j-1}, i_j - 1)} = \bigcup_{i_0 \leq n \leq i_p - 1} P_n \in$
> $\mathscr{K}^h$ for some $p \geq 2$, a strictly increasing sequence $(i_0, \dots, i_p)$ and $R_{(i_{j-1}, i_j - 1)} \in \mathscr{K}^{h+1}$ for all
> $1 \leq j \leq p$, then we simply sum $Vec$ over the indices from $i_0$ to $i_p - 1$. After that, the computed
> quantity is set in $newVec$ at index $i_0$. So actually computing $Succ_k$ is not needed and not done.

161

162 The computation time of the algorithm is in $O(m|\mathscr{K}|)$, which is fast for a single evaluation, but calling
163 it repeatedly on a path of selection sets $(S_t)_{t \in \mathbb{N}_m^*}$ has complexity $O(m^2|\mathscr{K}|)$ which is not desirable and
164 makes computations difficult in practice, hence the need for a new, faster algorithm.

165 *Remark 2.1.* The specific computation of the $R_k$'s and the $\zeta_k$'s such that Equation (6) holds is outside
166 the scope of the present paper, but different constructions can be found in Blanchard et al. (2020),

167 Durand et al. (2020) or Blain et al. (2022) for example.

## 3 New algorithms

### 3.1 Pruning the forest

170 We remark the simple fact that if, for example, $(1, 1), (2, 2), (1, 2) \in \mathcal{K}$, and $\zeta_{(1,2)} \geq \zeta_{(1,1)} + \zeta_{(2,2)}$, then
171 $R_{(1,2)}$ never contributes to the computation of any $V_{\mathfrak{R}}^*(S)$ and it could just be removed from $\mathfrak{R}$. We
172 now formalize and prove this pruning scheme.

173 **Definition 3.1** (Pruning). We define by $\mathcal{K}^{\mathfrak{pr}}$ ($\mathcal{K}$ pruned) the set of elements of $\mathcal{K}$ such that we
174 removed all $(i, i')$ such that there exists $p \geq 2$ and integers $i_1, \dots, i_{p-1}$ such that, when setting $i_0 = i$
175 and $i_p = i' + 1$, the sequence $(i_0, \dots, i_p)$ is strictly increasing, $(i_{j-1}, i_j - 1) \in \mathcal{K}$ for all $1 \leq j \leq p$ and
176 finally $\zeta_{(i,i')} = \zeta_{(i_0, i_p - 1)} \geq \sum_{j=1}^{p} \zeta_{(i_{j-1}, i_j - 1)}$.

177 An important note is that for a removed $(i, i') \in \mathcal{K} \setminus \mathcal{K}^{\mathfrak{pr}}$, we can always choose the indices $i_1, \dots, i_{p-1}$
178 such that actually $(i_j, i_{j+1} - 1) \in \mathcal{K}^{\mathfrak{pr}}$ and not only $\mathcal{K}$, because if $(i_j, i_{j+1} - 1) \in \mathcal{K} \setminus \mathcal{K}^{\mathfrak{pr}}$ it can itself
179 be fragmented, and this decreasing recursion eventually ends (the later possible being at the atoms
180 of the forest structure). Also note that removing elements from $\mathcal{K}$ does not alter the fact that we
181 have at hand a forest structure, that is, the reference family defined by $\mathfrak{R}^{\mathfrak{pr}} = (R_k, \zeta_k)_{k \in \mathcal{K}^{\mathfrak{pr}}}$ has a
182 forest structure. Because pruning a forest structure does not touch the atoms, note finally that if $\mathcal{K}$
183 is complete then so is $\mathcal{K}^{\mathfrak{pr}}$.

184 The following proposition states that pruning the forest does not alter the bound.

185 **Proposition 3.1.** *For any $S \subseteq \mathbb{N}_m^*$, $V_{\mathfrak{R}}^*(S) = V_{\mathfrak{R}^{\mathfrak{pr}}}^*(S)$.*

186 *Proof.* Recall Equation (11) and, because $\mathfrak{R}^{\mathfrak{pr}}$ also has a forest structure,

$$V_{\mathfrak{R}^{\mathfrak{pr}}}^*(S) = \min_{Q \subseteq \mathcal{K}^{\mathfrak{pr}}} \left( \sum_{k' \in Q} \zeta_{k'} \wedge |S \cap R_{k'}| + \left| S \setminus \bigcup_{k' \in Q} R_{k'} \right| \right), \tag{13}$$

187 so we immediately get that $V_{\mathfrak{R}}^*(S) \leq V_{\mathfrak{R}^{\mathfrak{pr}}}^*(S)$.

188 Let any $Q \subseteq \mathcal{K}$. We split $Q$ in $A$ elements of $\mathcal{K} \setminus \mathcal{K}^{\mathfrak{pr}}$, denoted $(i_{0,a}, i_{p_a,a} - 1)$, $1 \leq a \leq A$ for some $p_a \geq 2$,
189 and $B$ elements of $\mathcal{K}^{\mathfrak{pr}}$, simply denoted $k_b$, $1 \leq b \leq B$. By the definition of $\mathcal{K}^{\mathfrak{pr}}$ and the previous
190 remarks, for any $1 \leq a \leq A$, there exist integers $i_{1,a}, \dots, i_{p_a-1,a}$ such that $i_{0,a} < i_{1,a} < \cdots < i_{p_a-1,a} < i_{p_a,a}$,
191 $(i_{j-1,a}, i_{j,a} - 1) \in \mathcal{K}^{\mathfrak{pr}}$ for all $1 \leq j \leq p_a$, and $\zeta_{(i_{0,a}, i_{p_a,a} - 1)} \geq \sum_{j=1}^{p_a} \zeta_{(i_{j-1,a}, i_{j,a} - 1)}$. Now let

$$Q^{\mathfrak{pr}} = \{k_b : 1 \leq b \leq B\} \cup \{(i_{j-1,a}, i_{j,a} - 1) : 1 \leq a \leq A, 1 \leq j \leq p_a\}. \tag{14}$$

192 We have that $Q^{\mathfrak{pr}} \subseteq \mathcal{K}^{\mathfrak{pr}}$ and $\bigcup_{k \in Q} R_k = \bigcup_{k \in Q^{\mathfrak{pr}}} R_k$. Then,

$$\begin{aligned}
\sum_{k \in Q} \zeta_k \wedge |S \cap R_k| + \left| S \setminus \bigcup_{k \in Q} R_k \right| = {} & \sum_{b=1}^{B} \zeta_{k_b} \wedge |S \cap R_{k_b}| \\
& + \sum_{a=1}^{A} \zeta_{(i_{0,a}, i_{p_a,a} - 1)} \wedge |S \cap R_{(i_{0,a}, i_{p_a,a} - 1)}| \\
& + \left| S \setminus \bigcup_{k \in Q} R_k \right|,
\end{aligned}$$

but for all $1 \le a \le A$,

$$\zeta_{(i_{0,a}, i_{p_a,a}-1)} \ge \sum_{j=1}^{p_a} \zeta_{(i_{j-1,a}, i_{j,a}-1)}$$

$$\ge \sum_{j=1}^{p_a} \zeta_{(i_{j-1,a}, i_{j,a}-1)} \wedge |S \cap R_{(i_{j-1,a}, i_{j,a}-1)}|,$$

so the term $\sum_{a=1}^{A} \zeta_{(i_{0,a}, i_{p_a,a}-1)} \wedge |S \cap R_{(i_{0,a}, i_{p_a,a}-1)}|$ is greater than or equal to

$$\sum_{a=1}^{A} \left( \sum_{j=1}^{p_a} \zeta_{(i_{j-1,a}, i_{j,a}-1)} \wedge |S \cap R_{(i_{j-1,a}, i_{j,a}-1)}| \right) \wedge |S \cap R_{(i_{0,a}, i_{p_a,a}-1)}|,$$

which is simply equal to

$$\sum_{a=1}^{A} \sum_{j=1}^{p_a} \zeta_{(i_{j-1,a}, i_{j,a}-1)} \wedge |S \cap R_{(i_{j-1,a}, i_{j,a}-1)}|.$$

Furthermore $\left| S \setminus \bigcup_{k \in Q} R_k \right| = \left| S \setminus \bigcup_{k \in Q^{\mathfrak{pr}}} R_k \right|$ so finally:

$$\sum_{k \in Q} \zeta_k \wedge |S \cap R_k| + \left| S \setminus \bigcup_{k \in Q} R_k \right| \ge \sum_{k \in Q^{\mathfrak{pr}}} \zeta_k \wedge |S \cap R_k| + \left| S \setminus \bigcup_{k \in Q^{\mathfrak{pr}}} R_k \right| \tag{15}$$

$$\ge V^*_{\mathfrak{R}^{\mathfrak{pr}}}(S).$$

Note that Equation (15) is true even if there are some $b \in \{1, \dots, B\}, a \in \{1, \dots, A\}, j \in \{1, \dots, p_a\}$ such that $k_b = (i_{j-1,a}, i_{j,a} - 1)$. We minimize over all $Q$ to get that $V^*_{\mathfrak{R}}(S) \ge V^*_{\mathfrak{R}^{\mathfrak{pr}}}(S)$. □ □

This gives a practical way to speed up computations by first pruning the family before computing any $V^*_{\mathfrak{R}}(S)$, because $\mathscr{K}^{\mathfrak{pr}}$ is smaller than $\mathscr{K}$, and by the above Proposition there is no theoretical loss in doing so.

Furthermore, pruning can be done really simply by following Algorithm 1 for $S = \mathbb{N}^*_m$, and pruning when appropriate. This gives the following Algorithm 2 , assuming, for simplicity, that the family is complete. The computation time of the algorithm is the same as Algorithm 1 , that is $O(m|\mathscr{K}|)$. Note that the only differences between Algorithm 2 and Algorithm 1 are the pruning step and $\zeta_k$ replacing $\zeta_k \wedge |S \cap R_k|$, because $\zeta_k \le |R_k|$ and here $S = \mathbb{N}^*_m$, so $\zeta_k \wedge |\mathbb{N}^*_m \cap R_k| = \zeta_k$. Also note that the algorithm returns $V^*_{\mathfrak{R}}(\mathbb{N}^*_m)$ as a by-product. The following proposition states that Algorithm 2 indeed produces the pruned region as in Definition 3.1.

**Proposition 3.2.** *The final $\mathscr{L}$ returned by Algorithm 2 is equal to $\mathscr{K}^{\mathfrak{pr}}$: $\mathscr{L} = \mathscr{K}^{\mathfrak{pr}}$.*

*Proof.* First, $\mathscr{K} \setminus \mathscr{L} \subseteq \mathscr{K} \setminus \mathscr{K}^{\mathfrak{pr}}$ is trivial: a $k$ such that $\zeta_k \ge \sum_{k' \in Succ_k} Vec_{k'}$ obviously satisfies the condition of Definition 3.1 to be pruned.

Now let $(i, i') \in \mathscr{K} \setminus \mathscr{K}^{\mathfrak{pr}}$ an element that is pruned by Definition 3.1, so there exists $p \ge 2$ and integers $i_1, \dots, i_{p-1}$ such that, when setting $i_0 = i$ and $i_p = i' + 1$, the sequence $(i_0, \dots, i_p)$ is strictly increasing, $(i_{j-1}, i_j - 1) \in \mathscr{K}$ for all $1 \le j \le p$ and finally $\zeta_{(i,i')} = \zeta_{(i_0, i_p-1)} \ge \sum_{j=1}^{p} \zeta_{(i_{j-1}, i_j-1)}$. Then by the proof of Theorem 1 of Durand et al. (2020) but applied to $S = R_{(i,i')}$ we have that $\sum_{j=1}^{p} \zeta_{(i_{j-1}, i_j-1)} \ge \sum_{k' \in Succ_{(i,i')}} Vec_{k'}$ (see the unnumbered line just above Equation (A4) in that paper) and so $\zeta_{(i,i')} \ge \sum_{k' \in Succ_{(i,i')}} Vec_{k'}$ hence $(i, i')$ is pruned by Algorithm 2 and $\mathscr{K} \setminus \mathscr{K}^{\mathfrak{pr}} \subseteq \mathscr{K} \setminus \mathscr{L}$.

In the end, $\mathscr{K} \setminus \mathscr{K}^{\mathfrak{pr}} = \mathscr{K} \setminus \mathscr{L}$ so $\mathscr{K}^{\mathfrak{pr}} = \mathscr{L}$. □ □

**Algorithm 2** Pruning of $\mathfrak{R}$

1: **procedure** PRUNING($\mathfrak{R} = (R_k, \zeta_k)_{k \in \mathscr{K}}$ with $\mathfrak{R}$ complete)
2:   $\quad \mathscr{L} \leftarrow \mathscr{K}$
3:   $\quad H \leftarrow \max_{k \in \mathscr{K}} \phi(k)$                        ▷ maximum depth
4:   $\quad$ **for** $h = H - 1, \dots, 1$ **do**
5:   $\quad\quad \mathscr{K}^h \leftarrow \{k \in \mathscr{K} \,:\, \phi(k) = h\}$
6:   $\quad\quad newVec \leftarrow (0)_{k \in \mathscr{K}^h}$
7:   $\quad\quad$ **for** $k \in \mathscr{K}^h$ **do**
8:   $\quad\quad\quad Succ_k \leftarrow \{k' \in \mathscr{K}^{h+1} \,:\, R_{k'} \subseteq R_k\}$
9:   $\quad\quad\quad$ **if** $Succ_k = \emptyset$ **then**
10:  $\quad\quad\quad\quad newVec_k \leftarrow \zeta_k$
11:  $\quad\quad\quad$ **else**
12:  $\quad\quad\quad\quad$ **if** $\zeta_k \geq \sum_{k' \in Succ_k} Vec_{k'}$ **then**
13:  $\quad\quad\quad\quad\quad \mathscr{L} \leftarrow \mathscr{L} \setminus \{k\}$        ▷ pruning of the region indexed by $k$
14:  $\quad\quad\quad\quad$ **end if**
15:  $\quad\quad\quad\quad newVec_k \leftarrow \min\left(\zeta_k, \sum_{k' \in Succ_k} Vec_{k'}\right)$
16:  $\quad\quad\quad$ **end if**
17:  $\quad\quad$ **end for**
18:  $\quad\quad Vec \leftarrow newVec$
19:  $\quad$ **end for**
20:  $\quad$ **return** $(\mathscr{L}, \sum_{k \in \mathscr{K}^1} Vec_k)$
21: **end procedure**

## 3.2 Fast algorithm to compute a curve of confidence bounds on a path of selection sets

Let $(i_1, \dots, i_m)$ a permutation of $\mathbb{N}_m^*$, eventually random, and, for all $t \in \mathbb{N}_m^*$, let $S_t = \{i_1, \dots, i_t\}$ and $S_0 = \emptyset$. For example, $(i_1, \dots, i_m)$ can be the permutation ordering the $p$-values in increasing order and in that case $S_t$ becomes the set of indices of the $t$ smallest $p$-values. Assume that we want to compute all $V_{\mathfrak{R}}^*(S_t)$ for all $t \in \{0, \dots, m\}$, this is what we call the curve of confidence bounds indexed by $(i_1, \dots, i_m)$. Applying Algorithm 1 to compute $V_{\mathfrak{R}}^*(S_t)$ for a given $t$ has complexity $O(t|\mathscr{K}|)$, so using it to sequentially compute the full curve has complexity $O\left(\sum_{t=0}^m t|\mathscr{K}|\right) = O\left(m^2|\mathscr{K}|\right)$. In this section, we present a new algorithm that computes the curve with a $O(m|\mathscr{K}|)$ complexity. The algorithm will need that $\mathfrak{R}$ is complete, so if that is not the case we first need to complete $\mathfrak{R}$ following the Algorithm 2 of Durand et al. (2020), which has a $O(m|\mathscr{K}|)$ complexity. In the remainder of this section we assume that $\mathfrak{R}$ is complete.

We first recall and introduce some notation. Recall that $\phi$ is the depth function inside of $\mathfrak{R}$, that $\mathfrak{P} \subseteq \mathscr{P}(\mathscr{K})$ is the set of subsets of $\mathscr{K}$ that realize a partition, recall the important result stated by Equation (12), and that $\mathscr{K}^h = \{k \in \mathscr{K} \,:\, \phi(k) = h\}$ for all $1 \leq h \leq H$ where $H = \max_{k \in \mathscr{K}} \phi(k)$. For any $t \in \mathbb{N}_m^*$ and $1 \leq h \leq H$, we denote by $k^{(t,h)}$ the element of $\mathscr{K}^h$ such that $i_t \in R_{k^{(t,h)}}$ if it exists, and we denote by $h_{\max}(t)$ the highest $h$ such that $k^{(t,h)}$ exists.

**Example 3.1** (Continuation of Example 2.2 and Example 2.3). Assume that the reference family of Example 2.2 has been labeled as in Example 2.3 and completed. Let $(i_1, \dots, i_{25})$ such that $i_1 = 7$, $i_2 = 1$ and $i_3 = 24$. Then for $t = 1$, $k^{(t,1)} = (1,5)$, $k^{(t,2)} = (2,3)$, $k^{(t,3)} = (3,3)$ and $h_{\max}(t) = H = 3$. For $t = 2$, $k^{(t,1)} = (1,5)$, $k^{(t,2)} = (1,1)$, $k^{(t,3)}$ does not exist and $h_{\max}(t) = 2$. For $t = 3$, $k^{(t,1)} = (8,8)$, $k^{(t,2)}$ does not exist and $h_{\max}(t) = 1$.

Now we can finally present the new algorithm and the proof that it computes the curve $(V_{\mathfrak{R}}^*(S_t))_{t \in \mathbb{N}_m}$. We present two versions of the algorithm (strictly equivalent): one very formal ( Algorithm 3 ), to

introduce additional notation used in the proof of Theorem 3.1, and, later, a simpler version that is the one actually implemented ( Algorithm 4 ). Recall that a detailed illustration of the steps of the algorithms will be provided in Section 3.3.

---

**Algorithm 3** Formal computation of $(V^*_{\mathfrak{R}}(S_t))_{0 \le t \le m}$

---

1: **procedure** CURVE($\mathfrak{R} = (R_k, \zeta_k)_{k \in \mathscr{K}}$ with $\mathfrak{R}$ complete, path $(S_t)_{1 \le t \le m}$ with $S_t = \{i_1, \dots, i_t\}$)
2: $\quad \mathscr{P}^0 \leftarrow \{(i,i) : 1 \le i \le n\}$ $\qquad\qquad\qquad\qquad\qquad\qquad\qquad$ ▷ the set of all atoms indices
3: $\quad \mathscr{K}^-_0 \leftarrow \{k \in \mathscr{K} : \zeta_k = 0\}$
4: $\quad \eta^0_k \leftarrow 0$ for all $k \in \mathscr{K}$
5: $\quad$ **for** $t = 1, \dots, m$ **do**
6: $\quad\quad$ **if** $i_t \in \bigcup_{k \in \mathscr{K}^-_{t-1}} R_k$ **then**
7: $\quad\quad\quad \mathscr{P}^t \leftarrow \mathscr{P}^{t-1}$
8: $\quad\quad\quad \mathscr{K}^-_t \leftarrow \mathscr{K}^-_{t-1}$
9: $\quad\quad\quad \eta^t_k \leftarrow \eta^{t-1}_k$ for all $k \in \mathscr{K}$
10: $\quad\quad$ **else**
11: $\quad\quad\quad$ **for** $h = 1, \dots, h_{\max}(t)$ **do**
12: $\quad\quad\quad\quad \eta^t_{k^{(t,h)}} \leftarrow \eta^{t-1}_{k^{(t,h)}} + 1$
13: $\quad\quad\quad\quad$ **if** $\eta^t_{k^{(t,h)}} < \zeta_k$ **then**
14: $\quad\quad\quad\quad\quad$ Pass
15: $\quad\quad\quad\quad$ **else**
16: $\quad\quad\quad\quad\quad h^f_t \leftarrow h$ $\qquad\qquad\qquad\qquad\qquad\qquad\qquad\qquad$ ▷ final depth
17: $\quad\quad\quad\quad\quad \mathscr{P}^t \leftarrow \left( \mathscr{P}^{t-1} \setminus \{k \in \mathscr{P}^{t-1} : R_k \subseteq R_{k^{(t,h^f_t)}}\} \right) \cup \{k^{(t,h^f_t)}\}$
18: $\quad\quad\quad\quad\quad \mathscr{K}^-_t \leftarrow \mathscr{K}^-_{t-1} \cup \{k^{(t,h^f_t)}\}$
19: $\quad\quad\quad\quad\quad$ Break the loop
20: $\quad\quad\quad\quad$ **end if**
21: $\quad\quad\quad$ **end for**
22: $\quad\quad\quad$ **if** the loop has been broken **then**
23: $\quad\quad\quad\quad \eta^t_k \leftarrow \eta^{t-1}_k$ for all $k \in \mathscr{K}$ not visited during the loop, that is all $k \notin \{k^{(t,h)}, 1 \le h \le h^f_t\}$
24: $\quad\quad\quad$ **else**
25: $\quad\quad\quad\quad \mathscr{P}^t \leftarrow \mathscr{P}^{t-1}$
26: $\quad\quad\quad\quad \mathscr{K}^-_t \leftarrow \mathscr{K}^-_{t-1}$
27: $\quad\quad\quad\quad \eta^t_k \leftarrow \eta^{t-1}_k$ for all $k \in \mathscr{K}$ not visited during the loop, that is all $k \notin \{k^{(t,h)}, 1 \le h \le h_{\max}(t)\}$
28: $\quad\quad\quad$ **end if**
29: $\quad\quad$ **end if**
30: $\quad$ **end for**
31: $\quad$ **return** $\mathscr{P}^t, \eta^t_k$ for all $t = 1, \dots, m$ and $k \in \mathscr{K}$
32: **end procedure**

---

The core idea of the algorithm is that, as we increase $t$ and add new hypotheses in $S_t$, we inflate a counter $\eta^t_k$ for each region $R_k$, by 1 if $i_t \in R_k$ (line 12), by 0 if not (lines 23 and 27), but only until the counter reaches $\zeta_k$ (line 13). After this point, the hypotheses in $R_k$ don't contribute to $V^*_{\mathfrak{R}}(S_t)$, we keep track of those hypotheses with $\mathscr{K}^-_t$ (line 6), so as soon as $\eta^t_{k^{(t,h)}} = \zeta_k$ we update $\mathscr{K}^-_t$ by adding $k^{(t,h)}$ (line 18) to it and we update $\mathscr{P}^t$ accordingly (line 17).

We will see in the following Theorem 3.1 how this algorithm allows to compute $V^*_{\mathfrak{R}}(S_t)$. We first need a final notation. Let

$$\mathscr{K}_t = \{k \in \mathscr{K} : \exists k' \in \mathscr{P}^t : R_{k'} \subseteq R_k\}.$$

The elements of $\mathcal{K}_t$ index the regions of the forest that "are above'' the regions of the current partition-realizing $\mathscr{P}^t$. In particular, we always have, for any $t \in \mathbb{N}_m$, $\mathcal{K}^1 \subseteq \mathcal{K}_t$ and $\mathscr{P}^t \subseteq \mathcal{K}_t$. We can also remark that the sequence $(\mathcal{K}_t)_{0 \leq t \leq m}$ is non-increasing for the inclusion relation, and that $\mathcal{K}_0 = \mathcal{K}$.

**Theorem 3.1** (Fast curve computation). *Let any $t \in \mathbb{N}_m$. Then, $\mathscr{P}^t \in \mathfrak{P}$, and for all $k \in \mathcal{K}_t$, we have*

$$V_{\mathfrak{R}}^*(S_t \cap R_k) = \eta_k^t \tag{16}$$

*and*

$$V_{\mathfrak{R}}^*(S_t \cap R_k) = \sum_{\substack{k' \in \mathscr{P}^t \\ R_{k'} \subseteq R_k}} \zeta_{k'} \wedge |S_t \cap R_{k'}|. \tag{17}$$

*Furthermore,*

$$V_{\mathfrak{R}}^*(S_t) = \sum_{k \in \mathscr{P}^t} \zeta_k \wedge |S_t \cap R_k| = \sum_{k \in \mathcal{K}^1} \eta_k^t. \tag{18}$$

The proof of this Theorem is postponed to Section 3.4. The first equality of Equation (18) states that the minimum in (12) is realized on the partition $\mathscr{P}^t$, and the last equality of the same Equation is the basis of the following light corollary.

**Corollary 3.1** (Easy computation). *For $t \in \{0, \dots, m-1\}$, $V_{\mathfrak{R}}^*(S_{t+1}) = V_{\mathfrak{R}}^*(S_t)$ if $i_{t+1} \in \bigcup_{k \in \mathcal{K}_t^-} R_k$, and $V_{\mathfrak{R}}^*(S_{t+1}) = V_{\mathfrak{R}}^*(S_t) + 1$ if not.*

*Proof.* From (18), $V_{\mathfrak{R}}^*(S_{t+1}) = \sum_{k \in \mathcal{K}^1} \eta_k^{t+1}$ and $V_{\mathfrak{R}}^*(S_t) = \sum_{k \in \mathcal{K}^1} \eta_k^t$. If $i_{t+1} \in \bigcup_{k \in \mathcal{K}_t^-} R_k$, $\eta_k^{t+1} = \eta_k^t$ for all $k \in \mathcal{K}^1$. If not, $\eta_k^{t+1} = \eta_k^t$ for all $k \in \mathcal{K}^1$, $k \neq k^{(t+1,1)}$, whereas for $k = k^{(t+1,1)}$, $\eta_k^{t+1} = \eta_k^t + 1$. □ □

We note that, from Theorem 3.1 and Corollary 3.1, if one is only interested in the computation of the curve $\left(V_{\mathfrak{R}}^*(S_t)\right)_{1 \leq t \leq m}$, tracking $\mathscr{P}^t$ is actually useless, what is important is to track and update $\mathcal{K}_t^-$ correctly. Hence the simpler, alternative Algorithm 4 . Note that Algorithm 4 is less formal than Algorithm 3 : as in Algorithm 1 and Algorithm 2 , it recycles notation (mimicking the actual code implementation) so the $t$ subscript or superscript is dropped from the $\mathcal{K}_t^-$ and the $\eta_k^t$. In Algorithm 4 , the notation $V_t$ is actually equal to $V_{\mathfrak{R}}^*(S_t)$ by Corollary 3.1.

It is easy to see that each step $t$ has a complexity in $O(|\mathcal{K}|)$ hence the total complexity is in $O(m|\mathcal{K}|)$. This is because, if the regions are carefully stocked in memory, especially if their bounds (in terms of hypothesis index) are stocked, then finding $k^{(t,h)}$ has a complexity in $O(|\mathcal{K}^h|)$ and checking if $i_t \in \bigcup_{k \in \mathcal{K}_{t-1}^-} R_k$ has a complexity in $O(|\mathcal{K}|)$.

## 3.3 Illustration on a detailed example

We still continue Example 2.2 and Example 2.3. Recall that $m = 25$, $P_{1:5} = R_1 = \{1, \dots, 20\}$, $P_1 = R_2 = \{1, 2\}$, $P_{2:3} = R_3 = \{3, \dots, 10\}$, $P_{4:5} = R_4 = \{11, \dots, 20\}$, $P_2 = \{3, 4\}$, $P_3 = R_5 = \{5, \dots, 10\}$, $P_4 = R_6 = \{11, \dots, 16\}$, $P_5 = R_7 = \{17, \dots, 20\}$, $P_{6:7} = R_8 = \{21, 22\}$, $P_6 = \{21\}$, $P_7 = R_9 = \{22\}$ and $P_8 = \{23, 24, 25\}$.

Now assume that we have the following values for the $\zeta_k$'s: $\zeta_{(1,5)} = 6$, $\zeta_{(1,1)} = 2$, $\zeta_{(2,3)} = 1$, $\zeta_{(3,3)} = 4$, $\zeta_{(4,5)} = 4$, $\zeta_{(4,4)} = 2$, $\zeta_{(5,5)} = 3$, $\zeta_{(6,7)} = 2$, $\zeta_{(7,7)} = 0$. Because $P_2$, $P_6$ and $P_8$ come from the completion operation (see Section 2.3), we also have $\zeta_{(2,2)} = |P_2| = 2$, $\zeta_{(6,6)} = |P_6| = 1$ and $\zeta_{(8,8)} = |P_8| = 3$. Theses values are summarized in Figure 3.

We want to compute the curve $\left(V_{\mathfrak{R}}^*(S_t)\right)_{1 \leq t \leq 9}$ with $S_t = \{i_1, \dots, i_t\}$ and $i_1 = 11$, $i_2 = 17$, $i_3 = 12$, $i_4 = 13$, $i_5 = 18$, $i_6 = 3$, $i_7 = 19$, $i_8 = 22$ and $i_9 = 5$.

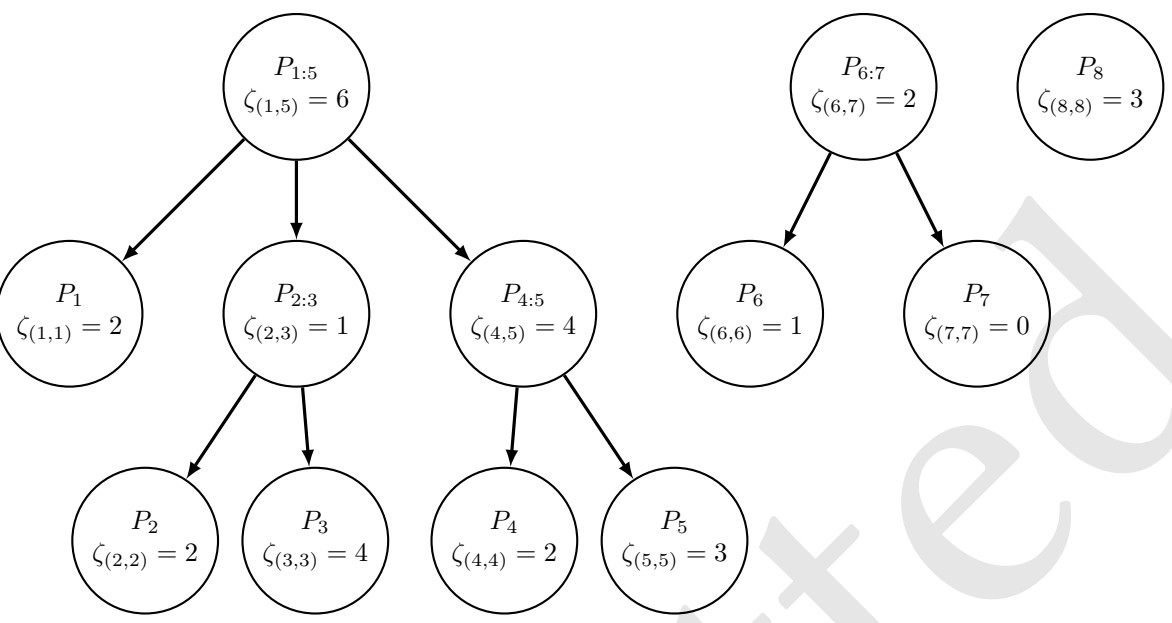

Figure 3: The regions of Example 2.2 with the $\zeta_k$ values.

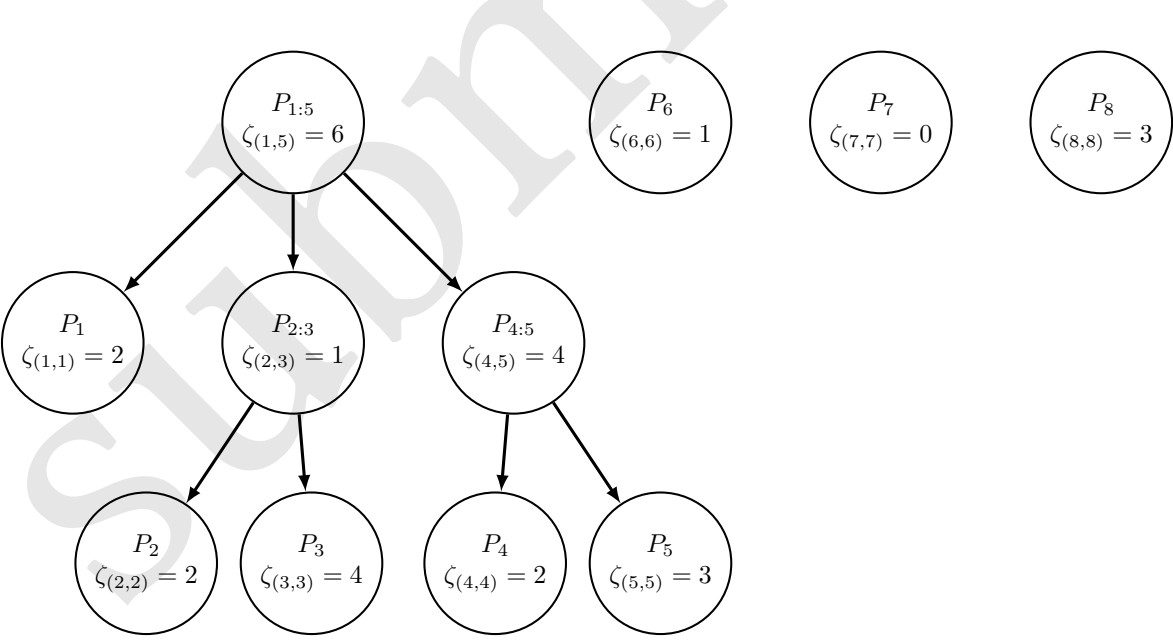

Figure 4: The regions of Example 2.2 after pruning.

**Algorithm 4** Practical computation of $(V_{\mathfrak{R}}^*(S_t))_{0 \leq t \leq m}$

1: **procedure** CURVE($\mathfrak{R} = (R_k, \zeta_k)_{k \in \mathscr{K}}$ with $\mathfrak{R}$ complete, path $(S_t)_{1 \leq t \leq m}$ with $S_t = \{i_1, \ldots, i_t\}$)
2:  $\quad V_0 \leftarrow 0$
3:  $\quad \mathscr{K}^- \leftarrow \{k \in \mathscr{K} : \zeta_k = 0\}$
4:  $\quad \eta_k \leftarrow 0$ for all $k \in \mathscr{K}$
5:  $\quad$ **for** $t = 1, \ldots, m$ **do**
6:  $\quad\quad$ **if** $i_t \in \bigcup_{k \in \mathscr{K}^-} R_k$ **then**
7:  $\quad\quad\quad V_t \leftarrow V_{t-1}$
8:  $\quad\quad$ **else**
9:  $\quad\quad\quad$ **for** $h = 1, \ldots, h_{\max}(t)$ **do**
10: $\quad\quad\quad\quad$ find $k^{(t,h)} \in \mathscr{K}^h$ such that $i_t \in R_{k^{(t,h)}}$
11: $\quad\quad\quad\quad \eta_{k^{(t,h)}} \leftarrow \eta_{k^{(t,h)}} + 1$
12: $\quad\quad\quad\quad$ **if** $\eta_{k^{(t,h)}} < \zeta_k$ **then**
13: $\quad\quad\quad\quad\quad$ pass
14: $\quad\quad\quad\quad$ **else**
15: $\quad\quad\quad\quad\quad \mathscr{K}^- \leftarrow \mathscr{K}^- \cup \{k^{(t,h)}\}$
16: $\quad\quad\quad\quad\quad$ break the loop
17: $\quad\quad\quad\quad$ **end if**
18: $\quad\quad\quad$ **end for**
19: $\quad\quad\quad V_t \leftarrow V_{t-1} + 1$
20: $\quad\quad$ **end if**
21: $\quad$ **end for**
22: $\quad$ **return** $(V_t)_{1 \leq t \leq m}$
23: **end procedure**

First, we apply Algorithm 2 to the family. This results in pruning $P_{6:7}$ (and only this region), because $2 = \zeta_{(6,7)} \geq \zeta_{(6,6)} + \zeta_{(7,7)} = 1 + 0$. This gives Figure 4.

Now we initialize Algorithm 3 , that is we let $t = 0$. Because $\zeta_{(7,7)} = 0$, $(7,7)$ is added to $\mathscr{K}_t^-$: $\mathscr{K}_0^- = \{(7,7)\}$. Furthermore, all $\eta_k^t$ are set to 0. The initial state of Algorithm 3 is shown in Figure 5, with $(7,7)$ being in red to show that it will not contribute to the computations.

We move on to $t = 1$, with $i_1 = 11$. $i_1 \in P_4 \subseteq P_{4:5} \subseteq P_{1:5}$. The appropriate $\eta_k^t$ are increased by one, and by (18) we have $V_{\mathfrak{R}}^*(S_1) = \eta_{(1,5)}^1 + \eta_{(6,6)}^1 + \eta_{(7,7)}^1 + \eta_{(8,8)}^1 = 1 + 0 + 0 + 0 = 1$. The state of the step is summarized in Figure 6.

We move on to $t = 2$, with $i_2 = 17$. $i_1 \in P_5 \subseteq P_{4:5} \subseteq P_{1:5}$. The appropriate $\eta_k^t$ are increased by one, and by (18) we have $V_{\mathfrak{R}}^*(S_2) = 2$. The state of the step is summarized in Figure 7.

We move on to $t = 3$, with $i_3 = 12$. $i_3 \in P_4 \subseteq P_{4:5} \subseteq P_{1:5}$. The appropriate $\eta_k^t$ are increased by one, and we notice that $\eta_{(4,4)}^3 = 2 = \zeta_{(4,4)}$. So $P_4$ will stop contributing, we add it to $\mathscr{K}_t^-$: $\mathscr{K}_3^- = \{(4,4),(7,7)\}$. By (18), we have $V_{\mathfrak{R}}^*(S_3) = 3$. The state of the step is summarized in Figure 8, with $P_4$ now also in red.

We move on to $t = 4$, with $i_4 = 13$. $i_4 \in P_4 \in \bigcup_{k \in \mathscr{K}_3^-} R_k$. No $\eta_k^t$ is increased (see line 9 of Algorithm 3 ), and by (18), we have $V_{\mathfrak{R}}^*(S_4) = 3$.

We move on to $t = 5$, with $i_5 = 18$. $i_5 \in P_5 \subseteq P_{4:5} \subseteq P_{1:5}$. We first increase $\eta_{(1,5)}^t$: $\eta_{(1,5)}^5 = 4 < \zeta_{(1,5)}$, then $\eta_{(4,5)}^t$: $\eta_{(4,5)}^5 = 4$, and we stop there because $\eta_{(4,5)}^5 = 4 = \zeta_{(4,5)}$. $P_{4:5}$ will stop contributing, we add it to $\mathscr{K}_t^-$: $\mathscr{K}_5^- = \{(4,5),(4,4),(7,7)\}$. Note that $\eta_{(5,5)}^t$ is not updated because we stopped the loop before, see line 23 of Algorithm 3 . By (18), we have $V_{\mathfrak{R}}^*(S_5) = 4$. The state of the step is summarized in Figure 9, with $P_{4:5}$ now also in red.

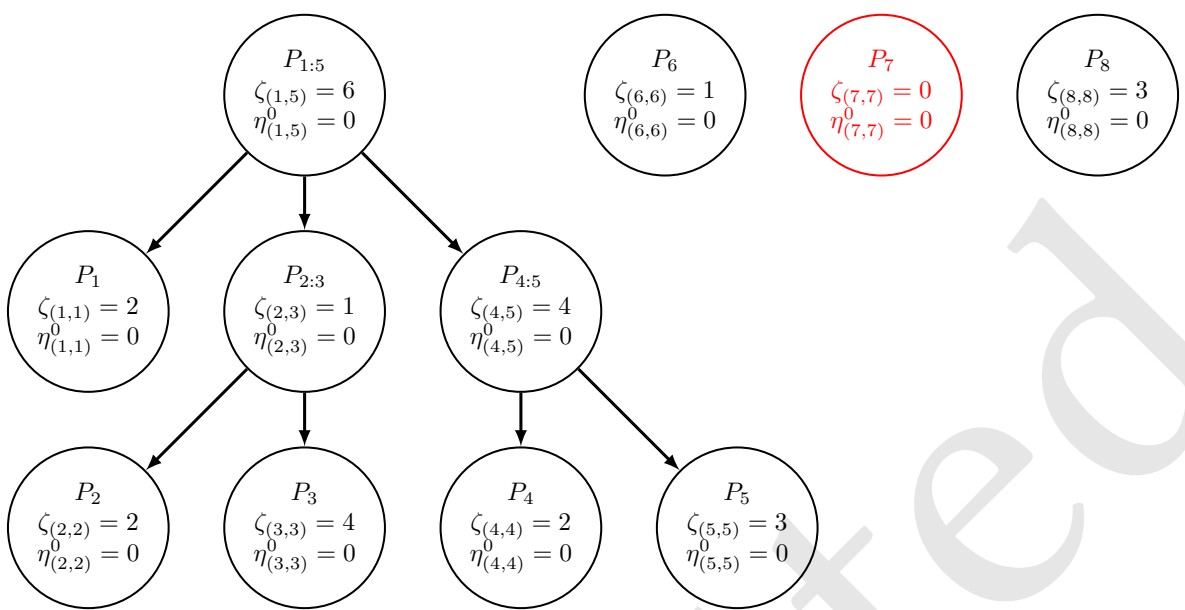

Figure 5: The regions of Example 2.2 at $t = 0$ in Algorithm 3 .

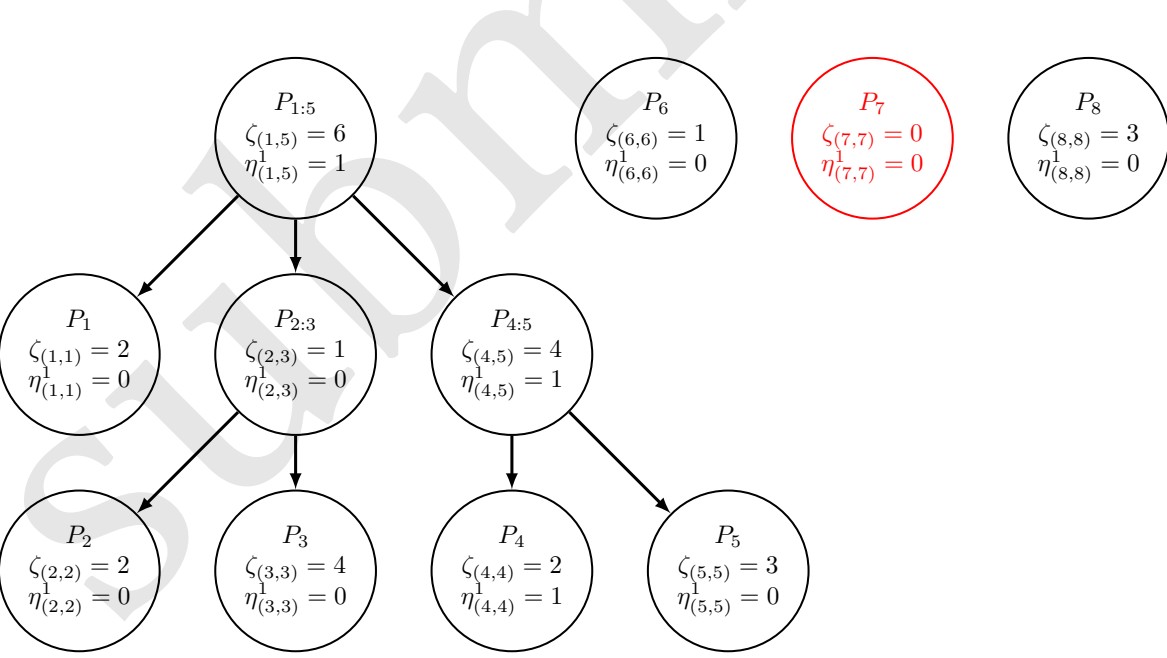

Figure 6: The regions of Example 2.2 at $t = 1$ in Algorithm 3 .

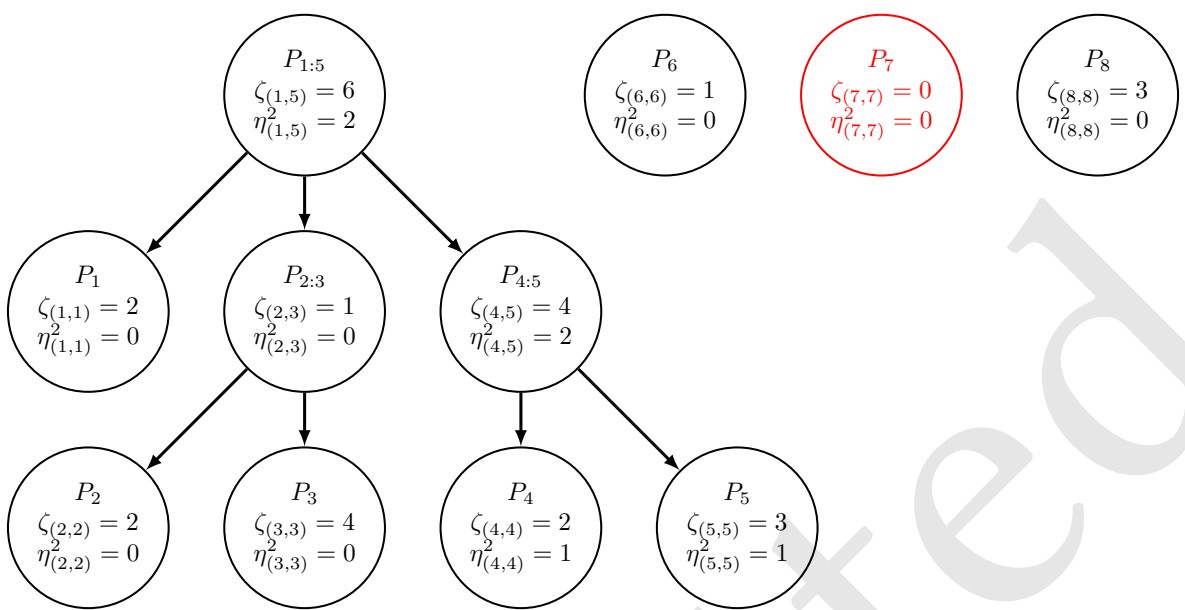

Figure 7: The regions of Example 2.2 at $t = 2$ in Algorithm 3 .

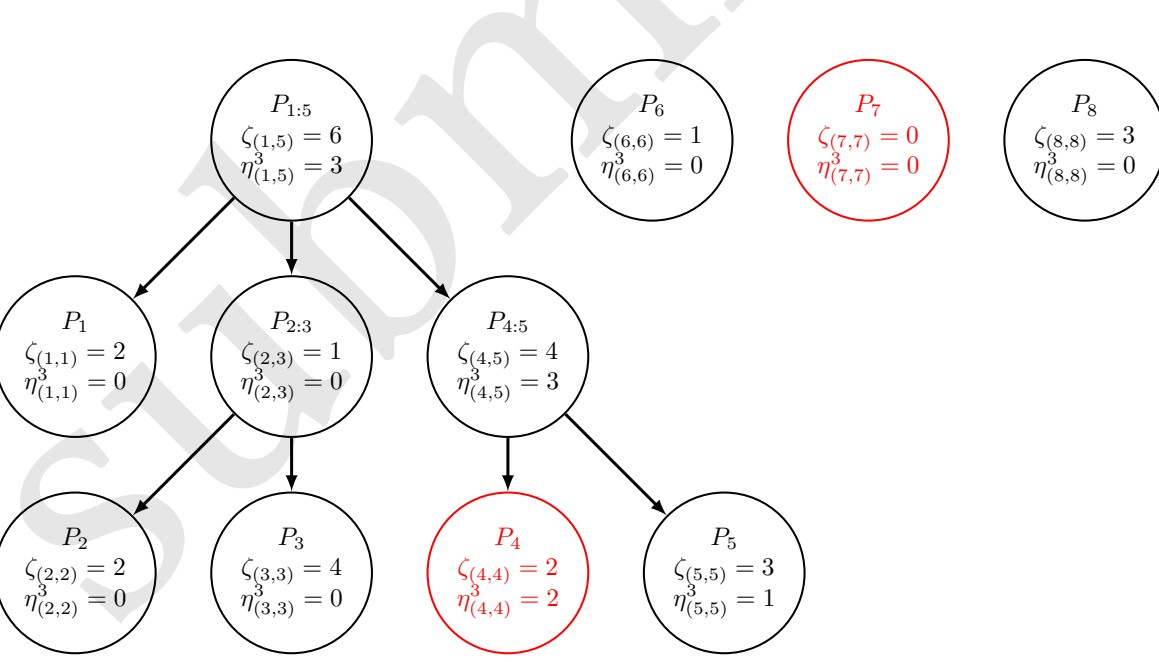

Figure 8: The regions of Example 2.2 at $t = 3$ in Algorithm 3 .

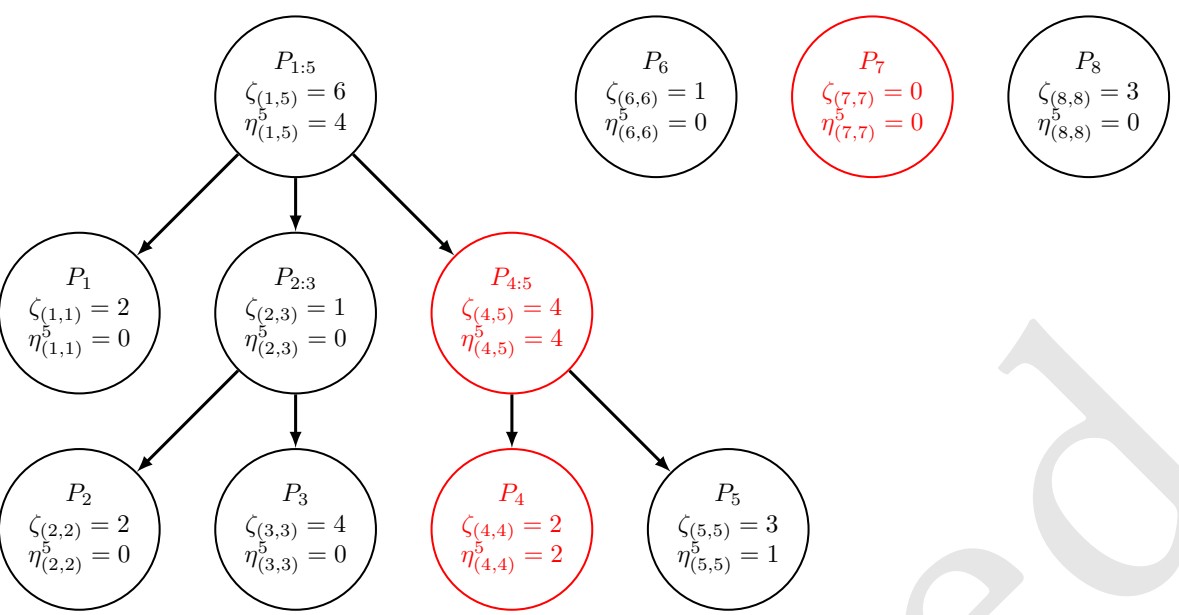

Figure 9: The regions of Example 2.2 at $t = 5$ in Algorithm 3 .

We move on to $t = 6$, with $i_6 = 3$. $i_6 \in P_3 \subseteq P_{2:3} \subseteq P_{1:5}$. We first increase $\eta^t_{(1,5)}$: $\eta^6_{(1,5)} = 5 < \zeta_{(1,5)}$, then $\eta^t_{(2,3)}$: $\eta^6_{(2,3)} = 1$, and we stop there because $\eta^6_{(2,3)} = 1 = \zeta_{(2,3)}$. $P_{2:3}$ will stop contributing, we add it to $\mathscr{K}^-_t$: $\mathscr{K}^-_6 = \{(2,3),(4,5),(4,4),(7,7)\}$. Note that $\eta^t_{(3,3)}$ is not updated because we stopped the loop before, see line 23 of Algorithm 3 . By (18), we have $V^*_{\mathfrak{R}}(S_6) = 5$. The state of the step is summarized in Figure 10, with $P_{2:3}$ now also in red.

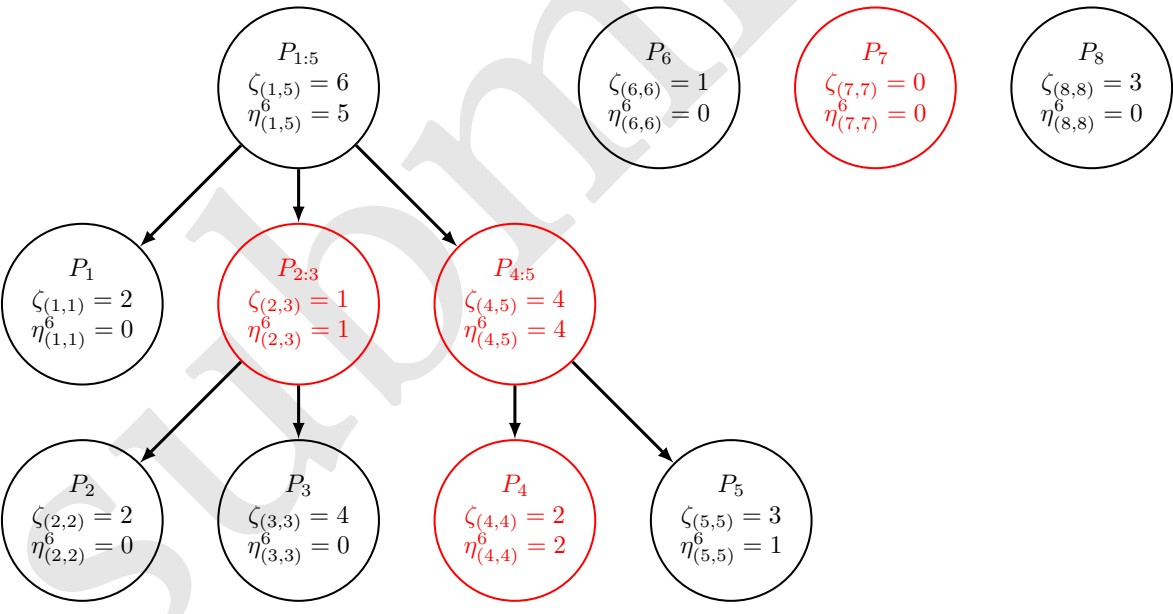

Figure 10: The regions of Example 2.2 at $t = 6$ in Algorithm 3 .

We move on to the remaining steps. $i_7 = 19 \in P_{4:5}$, $i_8 = 22 \in P_7$ and $i_9 = 5 \in P_{2:3}$ are all in $\bigcup_{k \in \mathscr{K}^-_6} R_k$ so no $\eta^t_k$ is increased at their step (see line 9 of Algorithm 3 ), and by (18), we have $V^*_{\mathfrak{R}}(S_7) = V^*_{\mathfrak{R}}(S_8) = V^*_{\mathfrak{R}}(S_9) = 5$.

### 3.4 Proof of Theorem 3.1

#### 3.4.1 Derivation of (18)

We first derive (18) from (16) and (17). First note that for all $Q \in \mathfrak{P}$,

$$Q = \bigcup_{k \in \mathscr{K}^1} \{k' \in Q \,:\, R_{k'} \subseteq R_k\} \tag{19}$$

and the union is disjoint. From (12), let $Q^* \in \mathfrak{P}$ such that $V_{\mathfrak{R}}^*(S_t) = \sum_{k' \in Q^*} \zeta_{k'} \wedge |S_t \cap R_{k'}|$. Then by (19),

$$
\begin{aligned}
V_{\mathfrak{R}}^*(S_t) &= \sum_{k' \in Q^*} \zeta_{k'} \wedge |S_t \cap R_{k'}| \\
&= \sum_{k \in \mathscr{K}^1} \sum_{\substack{k' \in Q^* \\ R_{k'} \subseteq R_k}} \zeta_{k'} \wedge |S_t \cap R_{k'}| \\
&= \sum_{k \in \mathscr{K}^1} \sum_{\substack{k' \in Q^* \\ R_{k'} \subseteq R_k}} \zeta_{k'} \wedge |S_t \cap (R_k \cap R_{k'})| \\
&= \sum_{k \in \mathscr{K}^1} \sum_{k' \in Q^*} \zeta_{k'} \wedge |(S_t \cap R_k) \cap R_{k'}| \tag{20} \\
&\geq \sum_{k \in \mathscr{K}^1} V_{\mathfrak{R}}^*(S_t \cap R_k), \tag{21}
\end{aligned}
$$

where the equality in (20) comes from the fact that if $R_{k'} \nsubseteq R_k$, then $R_{k'} \cap R_k = \varnothing$, that is, $R_k \subseteq R_{k'}$ is impossible because $k \in \mathscr{K}^1$. Furthermore, (21) holds again by (12).

Because $\mathscr{K}^1 \subseteq \mathscr{K}_t$, by (17), $V_{\mathfrak{R}}^*(S_t \cap R_k) = \sum_{\substack{k' \in \mathscr{P}^t \\ R_{k'} \subseteq R_k}} \zeta_{k'} \wedge |S_t \cap R_{k'}|$ for all $k \in \mathscr{K}^1$. Then,

$$
\begin{aligned}
\sum_{k \in \mathscr{K}^1} V_{\mathfrak{R}}^*(S_t \cap R_k) &= \sum_{k \in \mathscr{K}^1} \sum_{\substack{k' \in \mathscr{P}^t \\ R_{k'} \subseteq R_k}} \zeta_{k'} \wedge |S_t \cap R_{k'}| \\
&= \sum_{k \in \mathscr{P}^t} \zeta_k \wedge |S_t \cap R_k| \text{ by (19)} \\
&\geq V_{\mathfrak{R}}^*(S_t) \text{ by (12).}
\end{aligned}
$$

So we proved that $V_{\mathfrak{R}}^*(S_t) = \sum_{k \in \mathscr{P}^t} \zeta_k \wedge |S_t \cap R_k| = \sum_{k \in \mathscr{K}^1} V_{\mathfrak{R}}^*(S_t \cap R_k)$ and finally $V_{\mathfrak{R}}^*(S_t) = \sum_{k \in \mathscr{K}^1} V_{\mathfrak{R}}^*(S_t \cap R_k) = \sum_{k \in \mathscr{K}^1} \eta_k^t$ by (16), again because $\mathscr{K}^1 \subseteq \mathscr{K}_t$. Every equality in (18) is proven.

#### 3.4.2 Proof of (16) and (17)

We show the remainder of the statements by a strong recursion over $t$. We have $\mathscr{P}^0 \in \mathfrak{P}$ by definition, and given that $S_0 = \varnothing$ and $\eta_k^0 = 0$ for all $k \in \mathscr{K}$ (recall that $\mathscr{K}_0 = \mathscr{K}$), everything is equal to 0 in (16) and (17).

So we let $t \in \{0, \dots, m-1\}$, and assume that $\mathscr{P}^{t'} \in \mathfrak{P}$ and that (16) and (17) hold for all $t' \leq t$. In all the following, $\bar{k}$ is the element of $\mathscr{P}^t$ such that $i_{t+1} \in R_{\bar{k}}$. We will distinguish two cases: if $i_{t+1} \in \bigcup_{k \in \mathscr{K}_t^-} R_k$ or not. First we show an inequality that will be used in both cases. We have, for all $k \in \mathscr{K}_t$,

$$V_{\mathfrak{R}}^*(S_{t+1} \cap R_k) \leq \sum_{\substack{k' \in \mathscr{P}^t \\ R_{k'} \subseteq R_k}} \zeta_{k'} \wedge |S_{t+1} \cap R_{k'}|. \tag{22}$$

Indeed, by (12),

$$V_{\mathfrak{R}}^*(S_{t+1} \cap R_k) \leq \sum_{k' \in \mathscr{P}^t} \zeta_{k'} \wedge |S_{t+1} \cap R_k \cap R_{k'}|.$$

For any $k' \in \mathscr{P}^t$, we have either $R_{k'} \cap R_k = \emptyset$, in which case $|S_{t+1} \cap R_k \cap R_{k'}| = 0$, either $R_{k'} \subseteq R_k$, in which case $|S_{t+1} \cap R_k \cap R_{k'}| = |S_{t+1} \cap R_{k'}|$, but $R_k \subsetneq R_{k'}$ is impossible. Indeed, by definition of $\mathscr{K}_t$, there exists $\tilde{k} \in \mathscr{P}^t$ such that $R_{\tilde{k}} \subseteq R_k$, so $R_k \subsetneq R_{k'}$ would entail $R_{\tilde{k}} \subsetneq R_{k'}$ which is impossible since $k', \tilde{k} \in \mathscr{P}^t \in \mathfrak{P}$ and so $R_{\tilde{k}}$ and $R_{k'}$ are part of a partition of $\mathbb{N}_m^*$. This gives (22).

### 3.4.2.1 First case: $i_{t+1} \in \bigcup_{k \in \mathscr{K}_t^-} R_k$

In this case, $\mathscr{P}^{t+1} = \mathscr{P}^t \in \mathfrak{P}$ and $\mathscr{K}_{t+1} = \mathscr{K}_t$. For any $k \in \mathscr{K}_{t+1}$ such that $i_{t+1} \notin R_k$ (or, equivalently, such that $S_{t+1} \cap R_k = S_t \cap R_k$),

$$\sum_{\substack{k' \in \mathscr{P}^{t+1} \\ R_{k'} \subseteq R_k}} \zeta_{k'} \wedge |S_{t+1} \cap R_{k'}| = \sum_{\substack{k' \in \mathscr{P}^t \\ R_{k'} \subseteq R_k}} \zeta_{k'} \wedge |S_t \cap R_{k'}|$$

$$= V_{\mathfrak{R}}^*(S_t \cap R_k) \text{ by (17)}$$
$$= \eta_k^t \text{ by (16)}$$
$$= \eta_k^{t+1}$$

because $\eta_k^t = \eta_k^{t+1}$ for all $k \in \mathscr{K}$. Furthermore $S_{t+1} \cap R_k = S_t \cap R_k$ so $V_{\mathfrak{R}}^*(S_{t+1} \cap R_k) = V_{\mathfrak{R}}^*(S_t \cap R_k)$. So everything is proved for such a $k$.

Now we let $k \in \mathscr{K}_{t+1}$ such that $i_{t+1} \in R_k$ or, equivalently, such that $R_{\bar{k}} \subseteq R_k$. We first need to show that $\zeta_{\bar{k}} \leq |S_t \cap R_{\bar{k}}|$, and for that we need to distinguish two subcases: if $\bar{k}$ has been added to $\mathscr{P}^t$ during a previous step of the algorithm, of if not.

#### 3.4.2.1.1 First subcase: $\bar{k}$ has never been added during the process of line 17

Then $\bar{k} \in \mathscr{P}^0$ and $R_{\bar{k}}$ is an atom, so $i_{t+1} \in \bigcup_{k' \in \mathscr{K}_t^-} R_{k'}$ implies that $R_{\bar{k}} \subseteq \bigcup_{k' \in \mathscr{K}_t^-} R_{k'}$ (because of the forest structure). Let $k'_{\max}$ such that

$$R_{k'_{\max}} = \max\{R_{k'} : k' \in \mathscr{K}_t^-, R_{\bar{k}} \subseteq R_{k'}\}$$

(this a maximum for the inclusion relation, and it is well defined thanks to the forest structure). By reductio ad absurdum we show that $k'_{\max} = \bar{k}$. If that wasn't the case, by the joint construction of $\mathscr{P}^t$ and $\mathscr{K}_t^-$ during the algorithm we would have $k'_{\max} \in \mathscr{P}^t$ and a contradiction with the fact that $\mathscr{P}^t \in \mathfrak{P}$: we can't have both $\bar{k} \in \mathscr{P}^t$ and $k'_{\max} \in \mathscr{P}^t$ if they are distinct. So $k'_{\max} = \bar{k}$, so $\bar{k} \in \mathscr{K}_t^-$, but it cannot have been added to $\mathscr{K}_t^-$ during a previous step of the algorithm, otherwise it would have been added to $\mathscr{P}^t$, too. Hence $\bar{k} \in \mathscr{K}_0^-$ which means that $\zeta_{\bar{k}} = 0$ and $\zeta_{\bar{k}} = 0 \leq |S_t \cap R_{\bar{k}}|$.

#### 3.4.2.1.2 Second subcase: $\bar{k}$ has been added to $\mathscr{P}^t$ at a previous step

Let $t' \leq t$ be this step. This means that $\bar{k} = k^{(t', h_{t'}^f)}$ and that at that step $\eta_{\bar{k}}^{t'} = \zeta_{\bar{k}}$. Indeed, the if condition in line 13 failed so $\eta_{\bar{k}}^{t'} \geq \zeta_{\bar{k}}$, but for all $t'' < t'$ we had $\eta_{\bar{k}}^{t''} \leq \zeta_{\bar{k}}$ which implies equality. Also $\bar{k} \in \mathscr{P}^{t'}$ so $\bar{k} \in \mathscr{K}_{t'}$ so we can write

$$\zeta_{\bar{k}} = \eta_{\bar{k}}^{t'}$$
$$= V_{\mathfrak{R}}^*(S_{t'} \cap R_{\bar{k}}) \text{ by (16)}$$
$$\leq |S_{t'} \cap R_{\bar{k}}|$$
$$\leq |S_t \cap R_{\bar{k}}|.$$

360  This concludes the two subcases dichotomy: $\zeta_{\bar{k}} \leq |S_t \cap R_{\bar{k}}|$ and we can go back to our $k \in \mathscr{K}_{t+1}$ such
361  that $i_{t+1} \in R_k$ and $R_{\bar{k}} \subseteq R_k$.

362  We write the following chain:

$$
\begin{aligned}
V_{\mathfrak{R}}^*(S_{t+1} \cap R_k) &\leq \sum_{\substack{k' \in \mathscr{P}^t \\ R_{k'} \subseteq R_k}} \zeta_{k'} \wedge |S_{t+1} \cap R_{k'}| \text{ by (22) and } \mathscr{K}_{t+1} \subseteq \mathscr{K}_t \\
&= \sum_{\substack{k' \in \mathscr{P}^t \\ R_{k'} \subseteq R_k \\ k' \neq \bar{k}}} \zeta_{k'} \wedge |S_{t+1} \cap R_{k'}| + \zeta_{\bar{k}} \wedge |S_{t+1} \cap R_{\bar{k}}| \\
&= \sum_{\substack{k' \in \mathscr{P}^t \\ R_{k'} \subseteq R_k \\ k' \neq \bar{k}}} \zeta_{k'} \wedge |S_t \cap R_{k'}| + \zeta_{\bar{k}} \wedge (|S_t \cap R_{\bar{k}}| + 1) \\
&= \sum_{\substack{k' \in \mathscr{P}^t \\ R_{k'} \subseteq R_k \\ k' \neq \bar{k}}} \zeta_{k'} \wedge |S_t \cap R_{k'}| + \zeta_{\bar{k}} \wedge |S_t \cap R_{\bar{k}}| \text{ because } \zeta_{\bar{k}} \leq |S_t \cap R_{\bar{k}}| \\
&= \sum_{\substack{k' \in \mathscr{P}^t \\ R_{k'} \subseteq R_k}} \zeta_{k'} \wedge |S_t \cap R_{k'}| \\
&= V_{\mathfrak{R}}^*(S_t \cap R_k) \text{ by (17)} \\
&= \eta_k^t \text{ by (16)} \\
&= \eta_k^{t+1}.
\end{aligned}
$$

363  But on the other hand, $S_t \subseteq S_{t+1}$ and so (12) also gives $V_{\mathfrak{R}}^*(S_t \cap R_k) \leq V_{\mathfrak{R}}^*(S_{t+1} \cap R_k)$ and so in the end
364  we have the desired outcome:

$$
V_{\mathfrak{R}}^*(S_{t+1} \cap R_k) = \eta_k^{t+1} = \sum_{\substack{k' \in \mathscr{P}^{t+1} \\ R_{k'} \subseteq R_k}} \zeta_{k'} \wedge |S_{t+1} \cap R_{k'}|,
$$

365  which concludes this first case.

366  **3.4.2.2   Second case: $i_{t+1} \notin \bigcup_{k \in \mathscr{K}_t^-} R_k$**

367  We first prove that $\mathscr{P}^{t+1} \in \mathfrak{P}$ whether it came form the adjustment in line 17 or not. If it didn't, it
368  stayed equal to $\mathscr{P}^t \in \mathfrak{P}$. If it did, we have

$$
\mathscr{P}^{t+1} = \left( \mathscr{P}^t \setminus \{k \in \mathscr{P}^t, R_k \subseteq R_{k^{(t+1, h_{t+1}^f)}}\} \right) \cup \{k^{(t+1, h_{t+1}^f)}\}. \tag{23}
$$

369  To prove that $\mathscr{P}^{t+1} \in \mathfrak{P}$ in that case, it suffices to prove there are no $k' \in \mathscr{P}^t$ such that $R_{k^{(t+1, h_{t+1}^f)}} \subsetneq R_{k'}$.
370  If it was the case, because of the strict inclusion, we would have $k' \notin \mathscr{P}^0$, so $k'$ would have been
371  added to $\mathscr{P}^{t'}$ at a previous step $t' \leq t$ of the algorithm, but in that case it would also have been added
372  to $\mathscr{K}_{t'}^- \subseteq \mathscr{K}_t^-$. So in the end we would have

$$
i_{t+1} \in R_{k^{(t+1, h_{t+1}^f)}} \subsetneq R_{k'} \subseteq \bigcup_{k \in \mathscr{K}_t^-} R_k
$$

373  which is a contradiction and so $\mathscr{P}^{t+1} \in \mathfrak{P}$.

374 Like in the first case, considering a $k \in \mathcal{K}_{t+1} \subseteq \mathcal{K}_t$ such that $i_{t+1} \notin R_k$ is not problematic, because in
375 that case $k$ is not visited at all by the algorithm at step $t + 1 : \eta_k^{t+1} = \eta_k^t$, $\{k' \in \mathcal{P}^{t+1} : R_{k'} \subseteq R_k\} =$
376 $\{k' \in \mathcal{P}^t : R_{k'} \subseteq R_k\}$, and for all $k' \in \mathcal{K}$ such that $R_{k'} \subseteq R_k$, $S_{t+1} \cap R_{k'} = S_t \cap R_{k'}$. Hence, from

$$V_{\mathfrak{R}}^*(S_t \cap R_k) = \eta_k^t = \sum_{\substack{k' \in \mathcal{P}^t \\ R_{k'} \subseteq R_k}} \zeta_{k'} \wedge |S_{t+1} \cap R_{k'}|,$$

377 we directly have

$$V_{\mathfrak{R}}^*(S_{t+1} \cap R_k) = \eta_k^{t+1} = \sum_{\substack{k' \in \mathcal{P}^{t+1} \\ R_{k'} \subseteq R_k}} \zeta_{k'} \wedge |S_{t+1} \cap R_{k'}|.$$

378 So we now focus on the $k \in \mathcal{K}_{t+1}$ such that $i_{t+1} \in R_k$. Note that for such $k$,

$$\eta_k^{t+1} = \eta_k^t + 1 = V_{\mathfrak{R}}^*(S_t \cap R_k) + 1 = \sum_{\substack{k' \in \mathcal{P}^t \\ R_{k'} \subseteq R_k}} \zeta_{k'} \wedge |S_t \cap R_{k'}| + 1$$

379 by construction, by (16) and by (17). Indeed, such a $k$ is equal to a $k^{(t+1,h)}$ with $h \leq h_{max}(t + 1)$, and
380 even $h \leq h_{t+1}^f$ if the latter exists.

381 Also, similarly to the first case, for all $k \in \mathcal{K}_{t+1}$ such that $i_{t+1} \in R_k$ (recall that this is equivalent to
382 $R_{\bar{k}} \subseteq R_k$), we can write:

$$
\begin{aligned}
V_{\mathfrak{R}}^*(S_{t+1} \cap R_k) &\leq \sum_{\substack{k' \in \mathcal{P}^t \\ R_{k'} \subseteq R_k}} \zeta_{k'} \wedge |S_{t+1} \cap R_{k'}| \text{ by (22) and } \mathcal{K}_{t+1} \subseteq \mathcal{K}_t \\
&= \sum_{\substack{k' \in \mathcal{P}^t \\ R_{k'} \subseteq R_k \\ k' \neq \bar{k}}} \zeta_{k'} \wedge |S_{t+1} \cap R_{k'}| + \zeta_{\bar{k}} \wedge |S_{t+1} \cap R_{\bar{k}}| \\
&= \sum_{\substack{k' \in \mathcal{P}^t \\ R_{k'} \subseteq R_k \\ k' \neq \bar{k}}} \zeta_{k'} \wedge |S_t \cap R_{k'}| + \zeta_{\bar{k}} \wedge (|S_t \cap R_{\bar{k}}| + 1) \\
&\leq \sum_{\substack{k' \in \mathcal{P}^t \\ R_{k'} \subseteq R_k \\ k' \neq \bar{k}}} \zeta_{k'} \wedge |S_t \cap R_{k'}| + \zeta_{\bar{k}} \wedge |S_t \cap R_{\bar{k}}| + 1 \\
&= \sum_{\substack{k' \in \mathcal{P}^t \\ R_{k'} \subseteq R_k}} \zeta_{k'} \wedge |S_t \cap R_{k'}| + 1 \\
&= V_{\mathfrak{R}}^*(S_t \cap R_k) + 1 \text{ by (17).}
\end{aligned}
\tag{24}
$$

383 Note that by the joint construction of $\mathcal{K}_t^-$ and $\mathcal{P}^t$ on lines 17 and 18, the fact that $i_{t+1} \notin \bigcup_{k \in \mathcal{K}_t^-} R_k$
384 implies that $\bar{k}$ is the index of an atom, so actually $h_{\max}(t + 1) = \phi(\bar{k})$, $\bar{k} = k^{(t+1,\phi(\bar{k}))}$ and the $R_k$,
385 $k \in \mathcal{K}_t$, such that $R_{\bar{k}} \subseteq R_k$ are nested and are exactly indexed by the $k^{(t+1,h)}$, $1 \leq h \leq \phi(\bar{k})$. We now
386 prove that for all of them, $V_{\mathfrak{R}}^*(S_{t+1} \cap R_k) \geq V_{\mathfrak{R}}^*(S_t \cap R_k) + 1$, which will be true in particular for the
387 ones that are in $\mathcal{K}_{t+1}$, given that $\mathcal{K}_{t+1} \subseteq \mathcal{K}_t$. We do that by constructing some sets $A_h$ with good
388 properties with a descending recursion on $h$, starting from $\phi(\bar{k})$. We only give the first two steps of
389 the construction, because every other step is exactly the same as the second one, which contains the
390 recursive arguments. We go back to the real definition of $V_{\mathfrak{R}}^*$ to do so, for any $S \subseteq \mathbb{N}_m$:

$$V_{\mathfrak{R}}^*(S) = \max_{\substack{A \subseteq \mathbb{N}_m \\ \forall k' \in \mathcal{K}, |A \cap R_{k'}| \leq \zeta_{k'}}} |A \cap S| = \max_{\substack{A \subseteq S \\ \forall k' \in \mathcal{K}, |A \cap R_{k'}| \leq \zeta_{k'}}} |A|. \tag{25}$$

By (25), we have that $V_\mathfrak{R}^*(S_t \cap R_{k(t+1,\phi(\bar{k}))}) = |A_{\phi(\bar{k})}|$ for a given $A_{\phi(\bar{k})} \subseteq S_t \cap R_{k(t+1,\phi(\bar{k}))}$ and such that $|A_{\phi(\bar{k})} \cap R_{k'}| \le \zeta_{k'}$ for all $k' \in \mathscr{K}$. Now for the second set, we construct $A_{\phi(\bar{k})-1}$. Note that $V_\mathfrak{R}^*(S_t \cap R_{k(t+1,\phi(\bar{k})-1)}) = |B|$ for some $B \subseteq S_t \cap R_{k(t+1,\phi(\bar{k})-1)}$ and such that $|B \cap R_{k'}| \le \zeta_{k'}$ for all $k' \in \mathscr{K}$. By reductio ad absurdum, if there are strictly less than $V_\mathfrak{R}^*(S_t \cap R_{k(t+1,\phi(\bar{k})-1)}) - V_\mathfrak{R}^*(S_t \cap R_{k(t+1,\phi(\bar{k}))}) = |B| - |A_{\phi(\bar{k})}|$ elements in $S_t \cap R_{k(t+1,\phi(\bar{k})-1)} \setminus S_t \cap R_{k(t+1,\phi(\bar{k}))}$, then $|B| + |S_t \cap R_{k(t+1,\phi(\bar{k}))}| - |S_t \cap R_{k(t+1,\phi(\bar{k})-1)}| > |A_{\phi(\bar{k})}| = V_\mathfrak{R}^*(S_t \cap R_{k(t+1,\phi(\bar{k}))})$. Given that $B \cup (S_t \cap R_{k(t+1,\phi(\bar{k}))}) \subseteq S_t \cap R_{k(t+1,\phi(\bar{k})-1)}$, this entails $|B \cap S_t \cap R_{k(t+1,\phi(\bar{k}))}| = |B| + |S_t \cap R_{k(t+1,\phi(\bar{k}))}| - |B \cup (S_t \cap R_{k(t+1,\phi(\bar{k}))})| > V_\mathfrak{R}^*(S_t \cap R_{k(t+1,\phi(\bar{k}))})$ which contradicts the maximality of $A_{\phi(\bar{k})}$ in (25).

So we construct $A_{\phi(\bar{k})-1}$ by taking the disjoint union of $A_{\phi(\bar{k})}$ and $V_\mathfrak{R}^*(S_t \cap R_{k(t+1,\phi(\bar{k})-1)}) - V_\mathfrak{R}^*(S_t \cap R_{k(t+1,\phi(\bar{k}))})$ elements of $S_t \cap R_{k(t+1,\phi(\bar{k})-1)} \setminus S_t \cap R_{k(t+1,\phi(\bar{k}))}$. We now establish the properties of $A_{\phi(\bar{k})-1}$. First, $A_{\phi(\bar{k})-1} \subseteq S_t \cap R_{k(t+1,\phi(\bar{k})-1)}$, and $|A_{\phi(\bar{k})-1}| = V_\mathfrak{R}^*(S_t \cap R_{k(t+1,\phi(\bar{k})-1)})$. For all $k' \in \mathscr{K}$ such that $R_{k(t+1,\phi(\bar{k})-1)} \cap R_{k'} = \varnothing$, we have $|A_{\phi(\bar{k})-1} \cap R_{k'}| = 0 \le \zeta_{k'}$. Furthermore,

$$|A_{\phi(\bar{k})-1} \cap R_{k(t+1,\phi(\bar{k}))}| = |A_{\phi(\bar{k})} \cap R_{k(t+1,\phi(\bar{k}))}|$$
$$\le \zeta_{k(t+1,\phi(\bar{k}))}$$

by construction of $A_{\phi(\bar{k})}$. Finally, for all $k'$ such that $R_{k(t+1,\phi(\bar{k})-1)} \subseteq R_{k'}$, $|A_{\phi(\bar{k})-1} \cap R_{k'}| = |A_{\phi(\bar{k})-1}| = V_\mathfrak{R}^*(S_t \cap R_{k(t+1,\phi(\bar{k})-1)}) = |B|$ with the previously defined $B$, in particular $|B \cap R_{k'}| \le \zeta_{k'}$, but given that $B \subseteq S_t \cap R_{k(t+1,\phi(\bar{k})-1)}$, $|B \cap R_{k'}| = |B|$. Wrapping all those equalities, it comes that $|A_{\phi(\bar{k})-1} \cap R_{k'}| \le \zeta_{k'}$. In the end, $|A_{\phi(\bar{k})-1} \cap R_{k'}| \le \zeta_{k'}$ for all $k' \in \mathscr{K}$, so $A_{\phi(\bar{k})-1}$ realizes the maximum in (25) for $S_t \cap R_{k(t+1,\phi(\bar{k})-1)}$.

By applying exactly the same method, we recursively construct a non-increasing sequence $A_{\phi(\bar{k})} \subseteq \cdots \subseteq A_1$ such that for all $\ell \in \{1, \ldots, \phi(\bar{k})\}$ and $k' \in \mathscr{K}$, $A_\ell \subseteq S_t \cap R_{k(t+1,\ell)}$, $V_\mathfrak{R}^*(S_t \cap R_{k(t+1,\ell)}) = |A_\ell|$, and $|A_\ell \cap R_{k'}| \le \zeta_{k'}$. Furthermore for $\ell' > \ell$, $A_\ell \setminus A_{\ell'} \subseteq S_t \cap R_{k(t+1,\ell)} \setminus S_t \cap R_{k(t+1,\ell')}$. Also note that the fact that $i_{t+1} \notin \bigcup_{k \in \mathscr{K}_t^-} R_k$ implies that $\eta_{k(t+1,\ell)}^t < \zeta_{k(t+1,\ell)}$ for all $\ell \in \{1, \ldots, \phi(\bar{k})\}$. So by (16), $|A_\ell| < \zeta_{k(t+1,\ell)}$.

Let, for any $\ell \in \{1, \ldots, \phi(\bar{k})\}$, $\widetilde{A}_\ell = A_\ell \cup \{i_{t+1}\}$. Given that $A_\ell \subseteq S_t \cap R_{k(t+1,\ell)}$ and that $i_{t+1} \in S_{t+1} \setminus S_t$, $\widetilde{A}_\ell \subseteq S_{t+1} \cap R_{k(t+1,\ell)}$, $|\widetilde{A}_\ell| = |A_\ell| + 1$, and for all $\ell' \in \{1, \ldots, \phi(\bar{k})\}$, $|\widetilde{A}_\ell \cap R_{k(t+1,\ell')}| = |A_\ell \cap R_{k(t+1,\ell')}| + 1$. Note that if, furthermore, $\ell \ge \ell'$, then $A_\ell \subseteq A_{\ell'}$, so

$$|\widetilde{A}_\ell \cap R_{k(t+1,\ell')}| = |A_\ell \cap R_{k(t+1,\ell')}| + 1$$
$$\le |A_{\ell'} \cap R_{k(t+1,\ell')}| + 1$$
$$= |A_{\ell'}| + 1$$
$$< \zeta_{k(t+1,\ell')} + 1.$$

On the contrary, if $\ell < \ell'$, we write that

$$|\widetilde{A}_\ell \cap R_{k(t+1,\ell')}| = |A_\ell \cap R_{k(t+1,\ell')}| + 1$$
$$= |(A_\ell \setminus A_{\ell'}) \cap R_{k(t+1,\ell')}| + |A_{\ell'} \cap R_{k(t+1,\ell')}| + 1$$
$$< 0 + \zeta_{k(t+1,\ell')} + 1,$$

because $A_\ell \setminus A_{\ell'} \subseteq R_{k(t+1,\ell)} \setminus R_{k(t+1,\ell')}$ hence $(A_\ell \setminus A_{\ell'}) \cap R_{k(t+1,\ell')} = \varnothing$. In both cases, $|\widetilde{A}_\ell \cap R_{k(t+1,\ell')}| < \zeta_{k(t+1,\ell')} + 1$ so $|\widetilde{A}_\ell \cap R_{k(t+1,\ell')}| \le \zeta_{k(t+1,\ell')}$. Additionally, for all $k' \in \mathscr{K}$ such that $i_{t+1} \notin R_{k'}$, $|\widetilde{A}_\ell \cap R_{k'}| = |A_\ell \cap R_{k'}| \le \zeta_{k'}$.

In the end, $|\widetilde{A}_\ell \cap R_{k'}| \le \zeta_{k'}$ for all $k' \in \mathscr{K}$, so

$$V_\mathfrak{R}^*(S_{t+1} \cap R_{k(t+1,\ell)}) \ge |\widetilde{A}_\ell| \text{ by (25)}$$
$$= |A_\ell| + 1$$
$$= V_\mathfrak{R}^*(S_t \cap R_{k(t+1,\ell)}) + 1.$$

So, as we wanted, $V_{\mathfrak{R}}^*(S_{t+1} \cap R_k) \geq V_{\mathfrak{R}}^*(S_t \cap R_k) + 1$ for all $k \in \mathscr{K}_t$ such that $i_{t+1} \in R_k$ and so for all such $k$ that are in $\mathscr{K}_{t+1}$. So every inequality in (24) becomes an equality and we have proven that

$$V_{\mathfrak{R}}^*(S_{t+1} \cap R_k) = V_{\mathfrak{R}}^*(S_t \cap R_k) + 1 = \eta_k^t + 1 = \eta_k^{t+1},$$

that is, (16) is true at $t + 1$. Looking at the first line of (24), we also proved that

$$V_{\mathfrak{R}}^*(S_{t+1} \cap R_k) = \sum_{\substack{k' \in \mathscr{P}^t \\ R_{k'} \subseteq R_k}} \zeta_{k'} \wedge |S_{t+1} \cap R_{k'}|. \tag{26}$$

The only thing left to prove is that (26) is also true with $\mathscr{P}^{t+1}$ instead of $\mathscr{P}^t$, that is that (17) also holds at $t + 1$, or, put differently, that

$$\sum_{\substack{k' \in \mathscr{P}^t \\ R_{k'} \subseteq R_k}} \zeta_{k'} \wedge |S_{t+1} \cap R_{k'}| = \sum_{\substack{k' \in \mathscr{P}^{t+1} \\ R_{k'} \subseteq R_k}} \zeta_{k'} \wedge |S_{t+1} \cap R_{k'}|. \tag{27}$$

If $h_{t+1}^f$ does not exist, meaning that we didn't break the loop, $\mathscr{P}^{t+1} = \mathscr{P}^t$ so there is nothing to prove.

Now assume that $h_{t+1}^f$ exists. So (23) holds. We will split each term in (27) in a sum of two terms. First, note that by (23), for any $k' \in \mathscr{K}$ such that $R_{k'} \cap R_{k^{(t+1,h_{t+1}^f)}} = \emptyset$, we have that $k' \in \mathscr{P}^{t+1}$ if and only if $k' \in \mathscr{P}^t$. And so,

$$
\sum_{\substack{k' \in \mathscr{P}^{t+1} \\ R_{k'} \subseteq R_k}} \zeta_{k'} \wedge |S_{t+1} \cap R_{k'}| = \sum_{\substack{k' \in \mathscr{P}^{t+1} \\ R_{k'} \cap R_{k^{(t+1,h_{t+1}^f)}} = \emptyset \\ R_{k'} \subseteq R_k}} \zeta_{k'} \wedge |S_{t+1} \cap R_{k'}| + \zeta_{k^{(t+1,h_{t+1}^f)}} \wedge |S_{t+1} \cap R_{k^{(t+1,h_{t+1}^f)}}|
$$
$$
= \sum_{\substack{k' \in \mathscr{P}^t \\ R_{k'} \cap R_{k^{(t+1,h_{t+1}^f)}} = \emptyset \\ R_{k'} \subseteq R_k}} \zeta_{k'} \wedge |S_{t+1} \cap R_{k'}| + \zeta_{k^{(t+1,h_{t+1}^f)}} \wedge |S_{t+1} \cap R_{k^{(t+1,h_{t+1}^f)}}|.
$$

Recall that we already proved that there is no $k' \in \mathscr{P}^t$ such that $R_{k^{(t+1,h_{t+1}^f)}} \subseteq R_{k'}$, so for any $k' \in \mathscr{P}^t$, either $R_{k'} \cap R_{k^{(t+1,h_{t+1}^f)}} = \emptyset$ or $R_{k'} \subseteq R_{k^{(t+1,h_{t+1}^f)}}$. Hence the split

$$
\sum_{\substack{k' \in \mathscr{P}^t \\ R_{k'} \subseteq R_k}} \zeta_{k'} \wedge |S_{t+1} \cap R_{k'}| = \sum_{\substack{k' \in \mathscr{P}^t \\ R_{k'} \cap R_{k^{(t+1,h_{t+1}^f)}} = \emptyset \\ R_{k'} \subseteq R_k}} \zeta_{k'} \wedge |S_{t+1} \cap R_{k'}| + \sum_{\substack{k' \in \mathscr{P}^t \\ R_{k'} \subseteq R_{k^{(t+1,h_{t+1}^f)}} \\ R_{k'} \subseteq R_k}} \zeta_{k'} \wedge |S_{t+1} \cap R_{k'}|
$$
$$
= \sum_{\substack{k' \in \mathscr{P}^t \\ R_{k'} \cap R_{k^{(t+1,h_{t+1}^f)}} = \emptyset \\ R_{k'} \subseteq R_k}} \zeta_{k'} \wedge |S_{t+1} \cap R_{k'}| + \sum_{\substack{k' \in \mathscr{P}^t \\ R_{k'} \subseteq R_{k^{(t+1,h_{t+1}^f)}}}} \zeta_{k'} \wedge |S_{t+1} \cap R_{k'}|,
$$

where the last equality comes from the fact that $R_{k^{(t+1,h_{t+1}^f)}} \subseteq R_k$, because $k \in \mathscr{K}_{t+1}$, $i_{t+1} \in R_k$, and $k^{(t+1,h_{t+1}^f)} \in \mathscr{P}^{t+1}$.

Given the two previously made splits, it remains to prove that

$$
\sum_{\substack{k' \in \mathscr{P}^t \\ R_{k'} \subseteq R_{k^{(t+1,h_{t+1}^f)}}}} \zeta_{k'} \wedge |S_{t+1} \cap R_{k'}| = \zeta_{k^{(t+1,h_{t+1}^f)}} \wedge |S_{t+1} \cap R_{k^{(t+1,h_{t+1}^f)}}|.
$$

433 Interestingly, this does not depend on $k$ anymore. By (26), the left-hand side is equal to $V_{\mathfrak{R}}^*(S_{t+1} \cap$

434 $R_{k^{(t+1,h_{t+1}^f)}})$. Because we are breaking the loop at step $h_{t+1}^f$, $\eta_{k^{(t+1,h_{t+1}^f)}}^{t+1} = \zeta_{k^{(t+1,h_{t+1}^f)}}$. Finally, because (16)

435 holds at $t + 1$, $\eta_{k^{(t+1,h_{t+1}^f)}}^{t+1} = V_{\mathfrak{R}}^*(S_{t+1} \cap R_{k^{(t+1,h_{t+1}^f)}})$. Wrapping all these assertions:

$$\sum_{\substack{k' \in \mathscr{P}^t \\ R_{k'} \subseteq R_{k^{(t+1,h_{t+1}^f)}}}} \zeta_{k'} \wedge |S_{t+1} \cap R_{k'}| = V_{\mathfrak{R}}^*(S_{t+1} \cap R_{k^{(t+1,h_{t+1}^f)}})$$

$$= V_{\mathfrak{R}}^*(S_{t+1} \cap R_{k^{(t+1,h_{t+1}^f)}}) \wedge |S_{t+1} \cap R_{k^{(t+1,h_{t+1}^f)}}|$$
$$= \eta_{k^{(t+1,h_{t+1}^f)}}^{t+1} \wedge |S_{t+1} \cap R_{k^{(t+1,h_{t+1}^f)}}|$$
$$= \zeta_{k^{(t+1,h_{t+1}^f)}} \wedge |S_{t+1} \cap R_{k^{(t+1,h_{t+1}^f)}}|,$$

436 which achieves the second case and so the proof. $\square$

# 4 Implementation

438 All algorithms discussed in this manuscript are already implemented in the R (R Core Team, 2024)
439 package sanssouci (Neuvial et al., 2024) which is available on GitHub (see the References for the link)
440 and is dedicated to the computation of confidence bounds for the number of false positives. It also
441 hosts the implementation of the methods described in Blanchard et al. (2020) and Enjalbert-Courrech
442 and Neuvial (2022). Algorithm 1 is implemented as the V.star function, Algorithm 2 is implemented
443 as the pruning function, and Algorithm 4 is implemented as the curve.V.star.forest.fast function
444 (whereas the curve.V.star.forest.naive function just repeatedly calls V.star). Note that the
445 pruning function has a delete.gaps option that speeds up the computation even more by removing
446 unnecessary gaps introduced in the data structure by the pruning operation, those gaps being due to
447 the specific structure that is used to store the information of $\mathscr{K}$.

448 Speaking of the data structure, we briefly describe it, with an example. We represent $(R_k)_{k \in \mathscr{K}}$ by two
449 lists, C and leaf_list. leaf_list is a list of vectors, where leaf_list[[i]] is the vector listing
450 the hypotheses in the atom $P_i$. C is a list of lists. For $1 \leq h \leq H$, C[[h]] lists the regions at depth $h$,
451 using the index bounds of the atoms they are composed of. That is, the elements of the list C[[h]]
452 are vectors of size two, and if there is k, i and j such that C[[h]][[k]] = c(i, j), it means that
453 $(i, j) \in \mathscr{K}$, or in other words that $P_{i:j}$ is one of the regions, and that $\phi((i, j)) = h$. We emphasize that
454 the 1D structure of the hypotheses has to be respected by the user as the current implementation
455 implicitly uses it: that is, $P_1$ has to contain the hypotheses labeled $1, 2, ..., p$, $P_2$ has to contain the
456 hypotheses labeled $p + 1, ...$, and so on. Also, the hypotheses have to be in non-decreasing order:
457 leaf_list[[1]] has to be equal to c(1, 2, 3, ..., p) and not, say, c(2, 1, 3, ..., p).

458 **Example 4.1** (Implementation of Example 2.3). For the reference family given in Example 2.2 and
459 completed in Example 2.3, $H = 3$. For $h = 1$, we have C[[1]][[1]] = c(1, 5), C[[1]][[2]] = c(6,
460 7), C[[1]][[3]] = c(8, 8). For $h = 2$, we have C[[2]][[1]] = c(1, 1), C[[2]][[2]] = c(2,
461 3), C[[2]][[3]] = c(4, 5), C[[2]][[4]] = c(6, 6), C[[2]][[5]] = c(7, 7). For $h = 3$, we
462 have C[[3]][[1]] = c(2, 2), C[[3]][[2]] = c(3, 3), C[[3]][[3]] = c(4, 4), C[[3]][[4]] =
463 c(5, 5).

464 And then for the atoms, we have leaf_list[[1]] = c(1, 2), leaf_list[[2]] = c(3, 4),
465 leaf_list[[3]] = c(5, 6, 7, 8, 9, 10),   leaf_list[[4]] = c(11, 12, 13, 14, 15, 16),
466 leaf_list[[5]] = c(17, 18, 19, 20), leaf_list[[6]] = 21, leaf_list[[7]] = 22 and finally
467 leaf_list[[8]] = c(23, 24, 25).

The functions `dyadic.from.leaf_list`, `dyadic.from.window.size`, and `dyadic.from.height` return the appropriate data structure to represent a $\mathcal{K}$ that can be described as a dyadic tree, based on some entry parameters that can be inferred from the names of the functions. The completion of a forest structure, mentioned in Section 2.3, is done by the `forest.completion` function. Finally, the $\zeta_k$'s are computed as in Durand et al. (2020) by the `zetas.tree` function with `method=zeta.DKWM`.

# 5 Numerical experiments

In this Section, we present some numerical experiments aiming to demonstrate the impact of the pruning of Algorithm 2 (using the `delete.gaps` option mentioned in Section 4) and of the fast Algorithm 4 , in terms of computation time, compared to the only previously available method to compute a curve of confidence bounds. As mentioned in Section 2.3 and Section 4, this naive method simply consisted in a `for` loop repeatedly applying Algorithm 1 .

To compare the computation time, we use the R package `microbenchmark` version 1.5.0 (Mersmann, 2024) with R version 4.4.0 (2024-04-24) and `sanssouci` version 0.13.0, on a MacBook Air M1 (2020) running macOS 15.1.1. The package `microbenchmark` allows to run code snippets a given number `n_repl` of times, and to compute summary statistics on the computation time. The script executing the computation can be found in the same repository as this manuscript.

Four scenarios are studied, all based on a common setting which we first describe. A number $m$ of hypotheses is tested. We use a reference family $(R_k, \zeta_k)$ such that the $R_k$'s have a forest structure of maximal depth $H = 10$. The graph of the inclusion relations between the $R_k$'s is a binary tree, hence there are $2^H - 1 = 1023$ $R_k$'s and in particular $2^{H-1} = 512$ atoms. $P$-values are generated in a gaussian one-sided fashion where $H_{0,i} = \{\mathcal{N}(\boldsymbol{\mu}, \mathrm{Id}) : \mu_i = 0\}$, $H_{1,i} = \{\mathcal{N}(\boldsymbol{\mu}, \mathrm{Id}) : \mu_i = 4\}$, and $p_i(X) = 1 - \Phi(X_i)$. $\mathcal{H}_1$ is comprised of the leafs 1, 5, 9 and 10, that is $\mathcal{H}_1 = P_1 \cup P_5 \cup P_9 \cup P_{10}$. For each scenario, the curve $\left(V_{\mathfrak{R}}^*(\{1, \ldots, t\})\right)_{t \in \mathbb{N}_m^*}$ is computed. For the experiments including pruning, the pruning is done once before the `n_repl` replications, to mimic the practice where pruning only needs to be done once and for all, while the user may be interested in computing multiple bounds and curves after that.

In scenarios 1 and 2, $m = 1024$ (so the atoms are of size 2), in scenarios 3 and 4, $m = 10240$ (so the atoms are of size 10). In scenarios 1 and 3, the $\zeta_k$'s are estimated trivially by $\zeta_k = |R_k|$, and in scenarios 2 and 4, they are computed as in Durand et al. (2020) with the DKWM inequality (Dvoretzky et al., 1956, ; Massart, 1990). Because of the size of $m$ and the poor performances of the naive approach, we set `n_repl=100` in scenarios 1 and 2 and `n_repl=10` only in scenario s 3 and 4. The differences between the scenarios are summarized in Table 1.

Table 1: Differences between the scenarios

| parameter | Scenario 1 | Scenario 2 | Scenario 3 | Scenario 4 |
|---|---|---|---|---|
| $m$ | 1024 | 1024 | 10240 | 10240 |
| zeta computation | trivial | DKWM | trivial | DKWM |
| `n_repl` | 100 | 100 | 10 | 10 |

For the trivial $\zeta_k$ computation of scenarios 1 and 3, the pruning obviously deletes all non-atom regions so $|\mathcal{K}^{\mathrm{pr}}| = 512$. Whereas, for the particular instance $\omega \in \Omega$ in the experiments, $|\mathcal{K}^{\mathrm{pr}}| = 541$ for scenario 2, and $|\mathcal{K}^{\mathrm{pr}}| = 573$ for scenario 4. Those results alone illustrate the benefits of pruning with respect to the reduction of the cardinality of the reference family: the regions above atoms with no signal (or no detectable signal in the trivial scenarios) are pruned. The fact that the regions above

atoms with detectable signal are not pruned means that they are relevant for the confidences bounds (which had already been demonstrated in the simulation study of Durand et al. (2020)).

The summary statistics of the computation time in each scenario are presented in Table 2, Table 3, Table 4, and Table 5, and they are also presented as boxplots in Figure 11. The time unit is the second (in logarithmic scale in the boxplots).

Table 2: Scenario 1

| expr | min | lq | mean | median | uq | max | neval |
|---|---|---|---|---|---|---|---|
| naive.not.pruned | 3.6924007 | 3.7943906 | 3.8521256 | 3.8386487 | 3.8780412 | 4.5247099 | 100 |
| naive.pruned | 3.2822354 | 3.4126177 | 3.4758338 | 3.4614076 | 3.5061541 | 3.8822089 | 100 |
| fast.not.pruned | 0.1332744 | 0.1367000 | 0.1383806 | 0.1385039 | 0.1392707 | 0.1768691 | 100 |
| fast.pruned | 0.0921422 | 0.0945472 | 0.0974025 | 0.0954231 | 0.0978687 | 0.1908498 | 100 |

Table 3: Scenario 2

| expr | min | lq | mean | median | uq | max | neval |
|---|---|---|---|---|---|---|---|
| naive.not.pruned | 3.7280744 | 3.8025695 | 3.8514710 | 3.8451367 | 3.8831009 | 4.1891831 | 100 |
| naive.pruned | 3.3556131 | 3.4533210 | 3.4926114 | 3.4906796 | 3.5182172 | 3.8501820 | 100 |
| fast.not.pruned | 0.1214844 | 0.1246071 | 0.1265674 | 0.1260760 | 0.1279640 | 0.1407320 | 100 |
| fast.pruned | 0.0815349 | 0.0827995 | 0.0841622 | 0.0835618 | 0.0851062 | 0.0896013 | 100 |

Table 4: Scenario 3

| expr | min | lq | mean | median | uq | max | neval |
|---|---|---|---|---|---|---|---|
| naive.not.pruned | 332.1856576 | 335.5148922 | 337.9856658 | 338.2432916 | 340.3329972 | 344.6255264 | 10 |
| naive.pruned | 328.3186707 | 329.3081834 | 332.1861199 | 331.4335773 | 333.3563651 | 338.7111614 | 10 |
| fast.not.pruned | 1.4881838 | 1.4966417 | 1.5066370 | 1.5078498 | 1.5151194 | 1.5217546 | 10 |
| fast.pruned | 0.9354581 | 0.9418174 | 0.9498806 | 0.9512573 | 0.9550453 | 0.9675895 | 10 |

Table 5: Scenario 4

| expr | min | lq | mean | median | uq | max | neval |
|---|---|---|---|---|---|---|---|
| naive.not.pruned | 331.0124665 | 335.6357519 | 349.7740812 | 337.6459728 | 342.1652204 | 401.4881647 | 10 |
| naive.pruned | 331.2567637 | 332.2215437 | 363.5822362 | 333.0651271 | 335.8347696 | 493.5124771 | 10 |
| fast.not.pruned | 1.3575818 | 1.3588461 | 1.3733567 | 1.3641336 | 1.3762178 | 1.4460291 | 10 |
| fast.pruned | 0.9287399 | 0.9441687 | 0.9551275 | 0.9520622 | 0.9624959 | 0.9972532 | 10 |

On each scenario, using the fast algorithm is much faster than the naive approach, while pruning always gives a slight improvement over not pruning.

Comparing scenarios 1 and 2 first, we see that, as expected, there is no significant change in computation time for naive.not.pruned, while naive.pruned is faster in scenario 1, given that we prune more. But, on the other hand, fast.not.pruned and fast.pruned are both faster in scenario 2, even if we prune less. This is because, for the regions with signal, said signal is detected and so

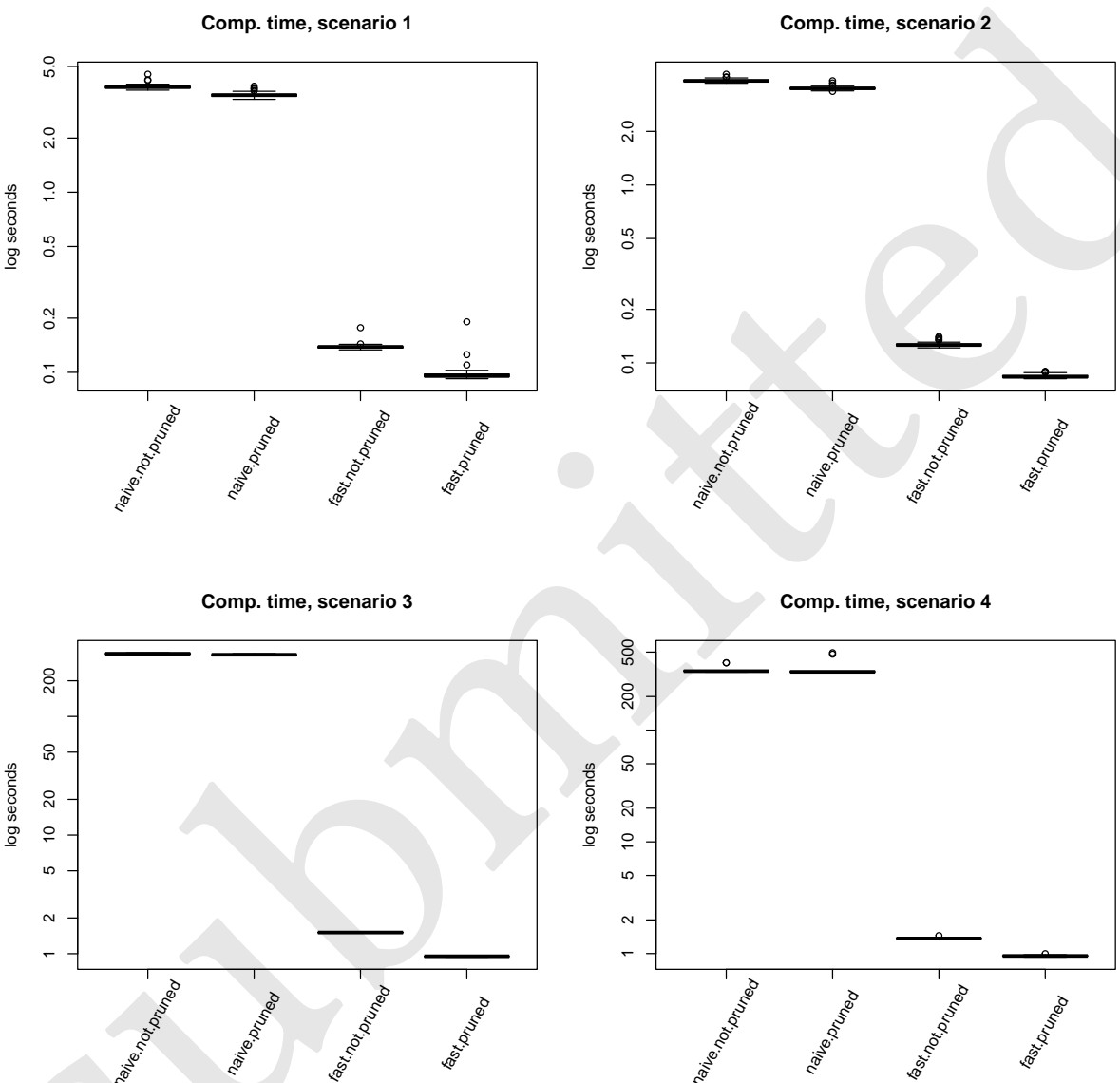

Figure 11: Computation times in scenario, in log seconds

those regions are quickly saturated, in the sense that we quickly have $\eta_k^t = \zeta_k$ and $k$ added to $\mathscr{K}_k^-$, which saves a lot of time.

The comparison between scenarios 3 and 4 is similar, except that this time we prune even less in scenario 4 and so the effect of the saturation is not enough to compensate. Although, with only n_repl=10, the statistics seem less accurate, this can be confirmed with additional experiments (n_repl can also be set to 100 without problem is we don't include naive methods).

Finally, comparing scenarios 3 & 4 with scenarios 1 & 2, we see that multiplying the number of hypotheses by 10 effectively multiplies the computation time by $\sim 10$ when using Algorithm 4 and by $\sim 100$ when using Algorithm 1 naively, which illustrates the stated complexities of $O(m|\mathscr{K}|)$ and $O(m^2|\mathscr{K}|)$, respectively.

# 6 Conclusion

In conclusion, we effectively introduced a new algorithm to compute a curve of confidence upper bounds, much faster the previous alternative, with one power of $m$ less in the complexity.

To develop new confidence upper bounds methodology and test them on simulations, it was previously not conceivable to replicate experiments a sufficient number of times while computing whole curves. For instance, in the simulation study of Durand et al. (2020), the number of replications chosen was 10 and the whole curve was not computed, only ten values along the curve were computed, for an m set to 12800, that is 0.078% of the curve had been computed. Now, simulation studies with an adequate number of replications and 100% of the curve become feasible.

A lot of work remains to be done on the sanssouci package. For example, to make the data format of a forest structure $(R_k)_{k\in\mathscr{K}}$ less convoluted and more user-friendly is an interesting project. Another one would be to implement inside the package the methods of the paper Blain et al. (2022), which are currently only available in the Python language (Van Rossum and Drake, 2009), and the methods of the paper Meah et al. (2024).

Other current works include the development of new reference families with theoretical JER control that could better account for realistic models, such as models with dependence between the $p$-values, see for example Perrot-Dockès et al. (2023), or models with discreteness.

# 7 Acknowledgements

This work has been supported by ANR-20-IDEES-0002 (PIA), ANR-19-CHIA-0021 (BISCOTTE), ANR-23-CE40-0018 (BACKUP) and ANR-21-CE23-0035 (ASCAI). Thanks to Romain Périer for being the first to extensively use the new implemented algorithms.

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

# Session information

```
R version 4.4.0 (2024-04-24)
Platform: aarch64-apple-darwin21.6.0
Running under: macOS 15.1.1

Matrix products: default
BLAS:   /opt/homebrew/Cellar/openblas/0.3.29/lib/libopenblasp-r0.3.29.dylib
LAPACK: /opt/homebrew/Cellar/r/4.4.0_1/lib/R/lib/libRlapack.dylib;  LAPACK version 3.12.0

locale:
[1] en_US.UTF-8/en_US.UTF-8/en_US.UTF-8/C/en_US.UTF-8/en_US.UTF-8

time zone: Europe/Paris
tzcode source: internal

attached base packages:
[1] stats     graphics  grDevices datasets  utils     methods   base

other attached packages:
[1] microbenchmark_1.5.0

loaded via a namespace (and not attached):
 [1] compiler_4.4.0   fastmap_1.1.1    cli_3.6.2        htmltools_0.5.8.1
 [5] tools_4.4.0      yaml_2.3.8       tinytex_0.51     rmarkdown_2.26
 [9] knitr_1.46       jsonlite_1.8.8   xfun_0.43        digest_0.6.35
[13] rlang_1.1.3      renv_1.0.7       evaluate_0.23
```

