# OpenReview forum: "A fast algorithm to compute a curve of confidence upper bounds for the False Discovery Proportion using a reference family with a forest structure"
_Computo — Accepted by Computo_

### Review · Reviewer_VFff · 2025-06-02

**Summary Of Contributions:**

The paper provides a useful computational strategy for an existing multiple testing principle, which provides simultaneous bounds for false discovery proportions. The proposed strategy is useful when the hypotheses are structured in a tree. Importantly, the paper has already been implemented in the R package “sanssouci”, which is a prominent R package implementing such methods.

The notation is somewhat involved but is correct as far as I’ve seen, and overall the theory seems correct, and the computational gains are illustrated with simulations.

**Audience:**

Yes

**Claims And Evidence:**

Yes

**Requested Changes:**

I have the following minor suggestions:

1.	In order to make the value of the paper clear to more readers, it would be good to mention some applications in e.g. the introduction.
2.	You use the notation “]0,1[”, however, “(0,1)” is much more common in statistics and mathematics so I would recommend that notation.
3.	Consider placing the proofs at the end.
4.	The title is very long. Consider e.g. “Fast confidence bounds for the false discovery proportion[…]”
5.	You sometimes have two squares behind a proof.

**Strengths And Weaknesses:**

See above. The focus is on improving the speed of an existing approach. This is relevant since these approaches are known to be slow when the number of hypotheses is large. The new method gives huge speed gains when there are many hypotheses.

I expect that the paper will only be studied by specialists, but importantly, the method has also been implemented in the prominent R package "sanssouci"; hence, the work can have a substantial scientific impact.

---

> ### Author Response · Authors · 2025-07-02
> **Response to reviewer VFff**
>
> Thank you a lot for your review. Here is my answer to all your points:
>
> 1. I have added two paragraphs in the introduction, one is about genomic applications, the other is about fMRI applications
> 2. I would like to keep ]0,1[, because I find the notation (0,1) to be ambiguous: (0,1) can be a 2-uplet of $\mathbb{N}^2$. If you insist, I will make the change, but I strongly prefer open brackets for open edges
> 3. I made a seventh section "Proofs" and moved all proofs in subsections there, except the proof of corollary 1, because this one takes only 2 lines.
> 4. Title shortened to "Fast confidence bounds for the false discovery proportion over a path of hypotheses"
> 5. As a result of 3., squares were not needed anymore, I deleted them all.
>
> In addition to your suggestions, I also reworked a lot my submission. Here is a summary of the main changes:
>
> 1. new implementation of the main algorithm, with the adequate complexity
> 2. new, refined complexity results for all algorithms of the paper
> 3. a new proposition about the relation between the main quantities appearing in the complexities
> 4. the numerical experiments were re-done with the new implementation and their discussion was updated accordingly
> 5. the pdf engine was switched to pdf-latex
> 6. correction of the definition of $\mathcal{P}^0$
> 7. correction of the proof of the first subcase in Theorem 1
> 8. simplification of the proof of the second subcase in Theorem 1
> 9. proof that $\mathcal{P}^0$ realizes a partition in Theorem 1
> 10. extension of the proof that $\mathcal{P}^t$ realizes a partition in Theorem 1
> 11. rigorous proof that $\zeta=\eta$ at the end of the proof of Theorem 1
> 12. addition of an R code snippet in the Implementation section
> 13. new tip discussing the (non-)need to complete a family before using the implementation of the algorithms
> 14. changed the detailed example of section 3.3 to make appear more diverse phenomenons
> 15. added the tracking of $\mathcal{P}^t$ in the detailed example of section 3.3
> 16. minor modifications and typo corrections
>
> All of this is in my submission as of commit 69b7c22fdbdb5f7d8e4d98483ab63e0e89500cbc (which, by coincidence, is my 300th commit): https://github.com/durandg12/algo-curve/tree/69b7c22fdbdb5f7d8e4d98483ab63e0e89500cbc

---

> ### Author Response · Authors · 2025-08-16
> **About (0,1)**
>
> I have changed my mind about the $(0,1)$ versus $]0,1[$ notation. I still prefer $]0,1[$, but it is true that $(0,1)$ has become widely used, even by my collaborators, and the potential amibiguity with the 2-uplet 0 and 1 is low in the context where I used the notation.

---

### Review · Reviewer_2dfZ · 2025-07-18

**Summary Of Contributions:**

This paper addresses the practical implementation of an algorithm for computing confidence bounds on the number of false positives within so-called selection sets. Constructing such bounds is particularly relevant in scientific settings involving multiple comparisons—for example, identifying brain regions associated with a specific physical task based on voxel-level analysis of fMRI data.

Confidence bounds in the context of selection can be obtained through various strategies. The approach adopted in this paper builds upon a line of research that leverages a reference family—a predefined collection of subsets—for which the number of false positives is controlled or known for each element. By exploiting subset relations between arbitrary selection sets and this reference family, one can derive valid confidence bounds for any selection set of interest.

However, the computation of bounds across a sequence of selection sets can be computationally intensive, with a time complexity that grows quadratically with the number of sets considered. This makes the development of more efficient algorithms crucial, particularly in high-dimensional applications.

The author focuses on a specific class of reference families—those exhibiting a forest structure—and demonstrates how this structure can be exploited to substantially accelerate computations. The key idea is to prune elements of the reference family that no longer contribute information about the number of false positives, thereby reducing unnecessary calculations.

Leveraging this insight, the authors introduce a new algorithm that achieves linear time complexity when computing bounds for a sequence of selection sets—matching the computational cost of computing a bound for a single set.

**Audience:**

Yes

**Claims And Evidence:**

Yes

**Requested Changes:**

Suggestions :

    - I suggest that the author add a small illustration of the first algorithm's steps (as done for Algorithm 4 in the figures).
	    The illustration can be more succinct, but it would help the reader quickly grasp the differences between the two algorithms.

    - In Algorithm 1, I wonder whether $Vec$ should be initialized with the $\zeta$ values of the sets at depth $H$.

    - I do not agree with the sentence "(we prefer to say upper bound instead of envelope for obvious reasons)" on line 29.
	    Using the term *envelope* can help emphasize the distinction between the functional $\hat{V}$ and the confidence bound for a single selection set.

    - I would recommend presenting the lower panel of Figure 11 at the beginning of the paper, as the improvement it shows is striking, yet currently only revealed at the very end.

------------------------------------------------------------------------------------------------------------------------------------------------------------------------------

List of typos (non-exhaustive) :

   - Line 21: "for exploratory analysis, like Genome ..." should be "for exploratory analysis, like in Genome ..."

   - Line 26: "notably advocated for in the context" should be "notably advocated in the context"

   - Line 45: Is there supposed to be a comma or a period between "closed testing" and the reference "Marcus et al."?

   - Line 173: "the set of elments of $\mathcal{K}$ such that"  consider rephrasing the part as "the set of elements of $\mathcal{K}$ from which..."

   - Line 528: "much faster the previous alternative" should be "much faster than the previous alternative"

**Strengths And Weaknesses:**

In my opinion, confidence bounds for selection sets are not only highly valuable in practice, but they also provide a ground for elegant and interesting mathematical developments. For this reason, I always enjoy reading papers on this topic.
The current manuscript presents substantial progress on the practical task of computing such confidence bounds. In particular, the proposed algorithm offers a significant improvement in computational efficiency, and the numerical experiments clearly support the theoretical claims regarding complexity.
My main criticism concerns the presentation of the results rather than their substance. I found the paper quite difficult to read, with a technical level that may be too high for the readership to follow. This made engagement with the content less enjoyable than it could have been, despite the quality of the scientific contribution.
To help improve the clarity of the exposition, I have provided some suggestions that I hope will enhance the accessibility and readability of the paper. I also include (non-exhaustive) minor comments regarding typographical errors.

---

> ### Author Response · Authors · 2025-07-18
> **Response to reviewer 2dfZ**
>
> Hello, thank you for your time and for the review. As I am about to leave for one week of holidays, here is a quick response to your points and suggestions. I will make the main changes in two weeks, when I get back to work.
>
> - I did not spend much time on Algorithm 1 because it is not new. In fact, it is one of the main results of the paper Durand, Blanchard, Neuvial & Roquain (2020). A step by step description of the algorithm is provided in that paper, in the last paragraph of Section 3 and in Figure 9. So, at your convenience, I can either refer to that description, or, as you suggested, write a new step by step description in the manuscript, using the same example as in the rest of the paper.
> - Thank you for this good catch, I indeed forgot to report the initialization of Algorithm 1 (which is in the third line of the version presented in Durand et al. (2020)). I will add this missing line.
> - For me and some collaborators, the term "envelope" may not refer to the functional aspect of the method (and for that, we sometimes rather use the term "simultaneous", as in a paper that we are currently writing), but to the existence of a lower bound in addition of the upper bound: that way, the number of false discoveries is "enveloped" between a lower and upper bound. But I totally agree that, sometimes, "envelope" indeed bears the "functional" meaning. Because of the possible ambiguity, I'd like to refrain to using it, but at the same time I deleted the sentence.
> - Thank you for the suggestion, I will add the scenario 3 figure in the Introduction.
> - I have corrected all the typos. I don't think there should be any punctuation between "closed tesing" and the Marcus et al. (1976), because this is the original reference about closed testing (to my knowledge).
>
> Please note that I made several other changes in the paper (including the correction of an error in the statement and in the proof of the main theorem), that I have listed in my response to the other reviewer. I hope some of those changes make the paper easier to grasp (for example, I have added explanations about what is $\mathcal{P}^t$).
>
> Thank you again. I will make another comment when I get back from holidays and I implement all the remaining changes.

---

> ### Author Response · Authors · 2025-08-16
> **Finishing the changes**
>
> Good morning.
>
> I have implemented the suggestions and remarks that were not already implemented:
>
> - I referred to the last paragraph of Section 3 and in Figure 9 of Durand et al. (2020) for a step by step description of Algorithm 1. I am still open to writing a new one in this manuscript if you think that is necessary.
> - I put back the missing initialization of Algorithm 1.
> - I changed my stance a little about the use of the term "envelope" after reviewing the literature again: it is now clear to me that the term "envelope" refers to the functional aspect of $V^*$.
> - I duplicated the figure of the numerical results of scenario 3 to put it at the end of the Introduction (and adapted the last paragraph to describe the figure).
> - given reviewer vH7P's remark and yours about the citation of Marcus et al., I put it in parentheses with a little sentence adding more context to the citation, that is, I precised that closed testing was introduced by Marcus et al. for FWER control.
>
> Thank you again, I will now begin to work on the requested changes by reviewer vH7P.

---

### Review · Reviewer_vH7P · 2025-07-24

**Summary Of Contributions:**

The authors provide a fast algorithm for the procedure of Durand et al. (2020), which computes simultaneous upper confidence for the False Discovery Proportion based on specific reference families with a forest structure. This new methodology is implemented in the R package sanssouci.

**Audience:**

Yes

**Broader Impact Concerns:**

None.

**Claims And Evidence:**

Yes

**Requested Changes:**

In addition to the other Reviewers’ points, I suggest improving the readability through the following minor changes:

* **Abstract:** Clearly state the old and new computational complexities, to highlight the computational gain.
* **Equation 1:** Some elements are either undefined or only briefly introduced at this point ($\mathcal{P}$, $\mathbb{P}$, the $m$ hypotheses $H_{0,i}$, the data $X$…). While these are later clarified in Section 2.1, which is well-organized and effective, I recommend either providing a more thorough preliminary description in the introduction or moving the formulae to Section 2.
* **Line 45:** I agree with Reviewer 2dfZ that the reference to Marcus et al. should be separated from the text or, preferably, placed in brackets.
* **Line 54:** Consider providing examples of this kind of structured hypotheses in practice.
* **Line 102:** Provide a short intuition of the concept of reference family and its defining elements.
* **Line 527:** I would underline again that the upper confidence bounds are for the FDP, as well as which settings the method is defined for.

**Strengths And Weaknesses:**

Inference on the FDP is gaining increasing attention in the field of multiple hypothesis testing. In practical applications, the main challenge lies in the computational burden, which can become prohibitive when testing a large number of null hypotheses, such as in genomics or neuroimaging. The proposed algorithm makes an existing methodology computationally feasible even in large-scale settings. This contribution is valuable and interesting due to both its theoretical foundations and practical applicability.

From my perspective, the only weakness of the manuscript is that it is technically dense and heavily relies on the frameworks established by Blanchard et al. (2020) and Durand et al. (2020). The extensive notation and frequent references to earlier work make the paper difficult to follow at times. However, I appreciated the recurring toy example and the related illustrations, which help convey the intuition behind some of the key concepts.

---

> ### Author Response · Authors · 2025-08-18
> **Response to reviewer vH7P**
>
> Thank you very much for your time and for your review. Here is a point by point answer to your reqyested change, in the same order as yours:
>
> - I have added a sentence with the old and new complexities in the abstract.
> - I expanded the introduction above and below Equation (1) to define more formally some elements, including $\mathcal P, \mathbb P, m, H_{0,i}, X$ and $\mathcal H_0$.
> - I put the Marcus et al. reference in parentheses with a little sentence adding more context to the citation, that is, I precised that closed testing was introduced by Marcus et al. for FWER control.
> - I expanded the introduction by giving 3 practical exemples where hypotheses can have the forest structure (genomic data, transcriptomic data, and brain imagery data).
> - I gave more intuition of the concept of reference families and its elements in two different places. First, I expanded the introduction with a couple of sentences describing the concept (which allowed me to go deeper with the concept of forest structure and then to answer the previous point). Second, I re-wrote a large part of Section 2.2 to insert a full parapgraph describing the intuition, after the definition.
> - I am sorry, but I didn't find the passage of the paper you were referring to. In the version that I submitted, line 527 is in the References, and none of the X27 lines seemed related to your requested change. I would gladly make the change if you can indicate me where it is.
>
> Please note that I made several other changes in the paper (including the correction of an error in the statement and in the proof of the main theorem), that I have listed in my response to reviewer VFff. I hope some of those changes make the paper easier to grasp.
>
> Thank you again.

---

> ### Comment · Reviewer_vH7P · 2025-08-21
> **Line 527**
>
> Thank you for your response. I referred to the first sentence in the Conclusions.

---

> > ### Author Response · Authors · 2025-08-21
> > **Line 527**
> >
> > Thank you, I have added your requested change at the beginning of the conclusion. So my last commit at https://github.com/durandg12/algo-curve now implements all requested changes by all reviewers. Unfortunately, there is still an issue in the computo github action call_env_workflow at the moment.

---

### Comment · Action_Editor_UxzK · 2025-02-10
**Page github non-accessible**

Bonjour,

Je suis l'éditeur associé responsable de votre manuscript. J'ai bien accès à votre dépôt github et au fichier PDF et HTML compilés. Par contre, la page github n'est pas activée et on ne peut donc pas lire le papier directement en ligne. Est-ce que vous pouvez activer la page github ?  Il s'agit de l'étape 2 décrite ici: https://computo.sfds.asso.fr/blog/2023/HTML-to-website/

Merci d'avance, Nelle Varoquaux, au nom du comité éditorial de Computo

---

> ### Author Response · Authors · 2025-02-10
> **Correction**
>
> Bonjour, c'est corrigé, j'ai effectué l'étape 2 décrite sur l'article de blog, j'ai aussi effectué l'étape 3. Elles ne sont malheureusement pas décrites dans le guide de soumission https://computo.sfds.asso.fr/submit/
>
> La page est à cette adresse : https://durandg12.github.io/algo-curve/, par contre j'y découvre un nouveau souci : les figures tikz ne sont pas générées, et sont remplacées par le code tikz. Je ne comprends pas, vu que sur ma machine le render produit un fichier HTML avec les figures (pareil pour le PDF).
>
> J'ai créé l'issue suivante : https://github.com/computorg/computo-quarto-extension/issues/37 pour remonter le problème.

---

> > ### Author Response · Authors · 2025-02-18
> > **Problème tikz réglé**
> >
> > Bonjour, le problème sur tikz que je mentionnais précédemment a l'air résolu, cf le lien de l'issue sur GitHub.

---

### Author Response · Authors · 2025-05-13
**2 issues**

I am currently solving two issues with my submission, see https://github.com/durandg12/algo-curve/issues/2 and https://github.com/durandg12/algo-curve/issues/3. I am in the process of solving them but I am a little delayed by other responsabilities. It will be done as soon as possible.

---

> ### Author Response · Authors · 2025-06-24
> **issues solved**
>
> Both issues are solved as of commit https://github.com/durandg12/algo-curve/commit/368169a1920d292498112ecf67719b284f3ef3f8

---

### Author Response · Authors · 2025-08-16
**Note to the editors and reviewers about a Computo issue**

Dear all, please note that due to a bug that I reported here: https://github.com/computorg/computo-quarto-extension/issues/42, the current version of my submission is not available at the URL https://durandg12.github.io/algo-curve/, an outdated version is shown instead.

To see the current version (as of the writing of this comment, I have implemented the requested changes by reviewers VFff and 2dfZ, but not yet vH7P), you have to download the repository and render the article on your local computer.

---

> ### Author Response · Authors · 2025-08-27
> **issue solved**
>
> The issue has been solved by last PR by Julien Chiquet. Thanks!

---

### Comment · Action_Editor_UxzK · 2025-09-22
**The production phase of your article is about to start**

Dear author,

Congratulations on the acceptance! Before the production phase of your paper starts, we need you to perform a couple of steps.

1. Can you please check that your affiliation and metadata are correctly filled in in the header of your article (name, affiliation, and url of each author).
2. Please run the CI and fix any remaining issues with the build.
3. Finally, transfer the ownership of the github repository to computo. See the [Authors guidelines](https://computo-journal.org/site/guidelines-authors.html#sec-acceptance) for a small description on how to perform the ownership transfer. In order to do so, you have been temporarily added to the computo organization.

Many thanks

---

> ### Author Response · Authors · 2025-09-23
> **Thank you**
>
> Hello, thank you very much for accepting and being the editor of my submission. And thanks again to the three reviewers.
>
> 1. I have verified the informations.
> 2. Unfortunately, there is a new issue in the build. I discovered it when I added a sentence to the Acknowledgements to thank the reviewers, four days ago. I have opened an issue here: https://github.com/computorg/computo-quarto-extension/issues/43. The error message is rather obscure:
> `FATAL (/opt/quarto/share/filters/main.lua:13615) An error occurred:
> This is an internal error. Please file a bug report at https://github.com/quarto-dev/quarto-cli/`
> 3. Should I wait for the resolution of the issue before transferring ownership, or should I do it right now? Will I still be able to submit pull requests to the manuscript if, one day, I find a typo?
>
> Thanks,
> Guillermo

---

> > ### Comment · Action_Editor_UxzK · 2025-09-24
> >
> > Hello Guillermo,
> >
> > Considering the error, I think it is best to transfer the ownership of the repository to us so that our technical team can have a look.
> >
> > Thanks,

---

> > > ### Author Response · Authors · 2025-09-24
> > >
> > > Hello, ok, I did the ownership transfer.
> > >
> > > Thanks

---

### Note · Reviewer_vH7P · 2025-09-02

**Comment:**

The author has addressed all the concerns raised during the review process.

**Audience:**

Yes

**Claims And Evidence:**

Yes

**Decision Recommendation:**

Accept

---

### Note · Reviewer_2dfZ · 2025-09-03

**Comment:**

The author have accounted with care all the suggestions made by the different reviewers. Most of the changes suggested were implemented by the author, if not the author provided convincing to argument in their way. The contribution presented in this work is valuable which is why I recommend accepting the paper.

**Audience:**

Yes

**Claims And Evidence:**

Yes

**Decision Recommendation:**

Accept

---

### Note · Reviewer_VFff · 2025-09-15

**Comment:**

I am satisfied with the changes the author has made based on my suggestions. I recommend acceptance.

**Audience:**

Yes

**Claims And Evidence:**

Yes

**Decision Recommendation:**

Accept

---

### Decision · Action_Editor_UxzK · 2025-09-22

**Recommendation:** Accept as is

**Comment:**

The authors have addressed all the points raised by the reviewers.

**Audience:**

This paper is relevant for computo's readership.

**Claims And Evidence:**

Dear authors,

I am pleased to inform you that your paper is accepted.

---

> ### Decision · Editors_In_Chief · 2025-09-22
>
> I approve the AE's decision.